# Graphs Help Graphs: Multi-Agent Graph Socialized Learning

**Jialu Li**[1,2,3]
jialuli@tju.edu.cn

**Yu Wang**[1,2,3,*]
wang.yu@tju.edu.cn

**Pengfei Zhu**[1,2,3,*]
zhupengfei@tju.edu.cn

**Wanyu Lin**[4]
wan-yu.lin@polyu.edu.hk

**Xinjie Yao**[1,2,3]
yaoxinjie@tju.edu.cn

**Qinghua Hu**[1,2,3]
huqinghua@tju.edu.cn

[1]College of Intelligence and Computing, Tianjin University, Tianjin, China
[2]Engineering Research Center of City Intelligence and Digital Governance,
Ministry of Education of the People's Republic of China, Tianjin, China
[3]Haihe Lab of ITAI, Tianjin, China
[4]Department of Computing, The Hong Kong Polytechnic University, Hong Kong, China

## Abstract

Graphs in the real world are fragmented and dynamic, lacking collaboration akin to that observed in human societies. Existing paradigms present collaborative information collapse and forgetting, making collaborative relationships poorly autonomous and interactive information insufficient. Moreover, collaborative information is prone to loss when the graph grows. Effective collaboration in heterogeneous dynamic graph environments becomes challenging. Inspired by social learning, this paper presents a Graph Socialized Learning (GSL) paradigm. We provide insights into graph socialization in GSL and boost the performance of agents through effective collaboration. It is crucial to determine with whom, what, and when to share and accumulate information for effective GSL. Thus, we propose the "Graphs Help Graphs" (GHG) method to solve these issues. Specifically, it uses a graph-driven organizational structure to select interacting agents and manage interaction strength autonomously. We produce customized synthetic graphs as an interactive medium based on the demand of agents, then apply the synthetic graphs to build prototypes in the life cycle to help select optimal parameters. We demonstrate the effectiveness of GHG in heterogeneous dynamic graphs by an extensive empirical study. The code is available through `https://github.com/Jillian555/GHG`.

## 1 Introduction

Graphs have become a powerful tool for capturing complex relationships between real-world entities, leading to their widespread use in social networks [1, 2], recommendation systems [3], and bioinformatics [4]. However, real-world graphs tend to be fragmented and dynamic, and there is often a lack of collaboration between different graphs. From the emergent behavior of ant colonies to the human collective intelligence, collaboration is common across different societies [5, 6]. It enables individuals to form social relationships and accomplish complex tasks. Inspired by social learning in human society, socialized learning has been extensively explored [7, 8]. It aims to have multiple agents interact, collaborate, and share knowledge. This acquired knowledge benefits other agents attempting to learn different yet related tasks, demonstrating capabilities beyond a single agent.

---

[*]Yu Wang and Pengfei Zhu are both the corresponding authors.

39th Conference on Neural Information Processing Systems (NeurIPS 2025).

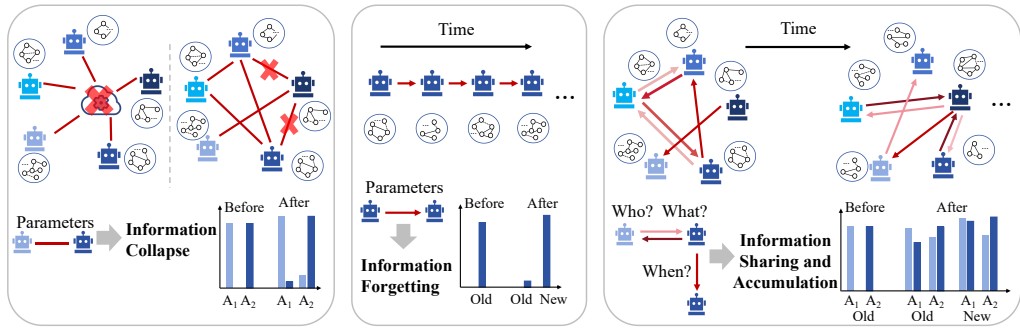

(a) Graph Federated Learning    (b) Graph Lifelong Learning    (c) Graph Socialized Learning

Figure 1: Comparison of different learning paradigms.

Existing paradigms strive to learn new graph knowledge collaboratively, just as human societies do. As shown in Figure 1, the star graph with a central server in graph centralized federated learning [9, 10, 11, 12] and the undirected topology in graph decentralized federated learning [13, 14, 15, 16]. Decentralized methods address communication bottlenecks and system crash risks of centralized methods [17]. Despite their effectiveness, on the one hand, they rely on undirected symmetric communication topologies, where agents fail to capture agent dependencies accurately. On the other hand, merely interacting with parameters may not lead to knowledge compatibility, causing information collapse and difficulty in obtaining capabilities from heterogeneous agents. Additionally, the graph lifelong learning paradigm [18, 19, 20, 21, 22] can adapt to dynamically growing graphs in the same parameter space for single-agent setups, but partial old task information is forgotten, causing agents to lose their initial performance as training continues. Thus, current paradigms are limited in achieving effective collaboration in heterogeneous dynamic graph environments.

Consider a real-world scenario: conducting collaborative training across growing citation networks on different platforms. Sharing elements like titles, author information, and citation relationships can boost the efficiency of academic paper retrieval. By re-visiting existing paradigms, we find that two significant challenges still exist: (1) Effectively managing graph heterogeneity among agents, thereby building autonomous collaborative relationships and sharing sufficient information to avoid collaborative information collapse. (2) Accurately capturing task dynamics in dynamic graphs to prevent the risk of forgetting accumulated collaborative information. Therefore, we raise the question: *1. How to realize effective collaboration in heterogeneous dynamic graph scenarios?*

As shown in Figure 1, we establish the Graph Socialized Learning (GSL) paradigm. It offers an effective way for agents to share and accumulate graph information in heterogeneous dynamic environments. This enables agents to learn the abilities of other agents and preserve old information without loss through effective collaboration. This exploration leads us to consider: *2. How to determine with whom, what, and when to share and accumulate graph information to accomplish effective GSL?*

Based on property and generalization of graph socialization, we propose the method "Graphs Help Graphs" (GHG) to accomplish GSL from three elements—**Who** (Organizational Structure): We employ graph-driven organizational structure to autonomously perceive collaborative relationship and strength among agents, thereby avoiding low-quality and redundant information from collaborators. Graph collaboration module models the collaboration graph by leveraging complementarity, as well as parameters (or features) and structure similarity between agents. **What** (Interactive Medium): We generate customized synthetic graphs as an interactive medium for information exchange. This enables the generation of the needed graphs based on agent states, thus addressing the issue of insufficient information from collaborators. A fine-grained customized collaboration evaluates the demand for different classes between agents. **When** (Life Cycle): We apply synthetic graphs to construct task prototypes, facilitating collaboration using optimal agent parameters at distinct tasks and preventing collaborative information from being forgotten. Our method achieves outstanding performance on seven datasets, surpassing the methods based on existing paradigms. Our main contributions can be summarized as follows:

- We present a practical learning paradigm called Graph Socialized Learning (GSL), enabling each agent's growth via collaborative interaction.

- Graph-driven organizational structure, customized interactive medium, and prototype-based life cycle form three key elements of socialized collaboration.

- Our method consistently achieves performance improvements on multiple datasets and demonstrates the effectiveness of all components.

## 2   Related Work

### 2.1   Graph Federated Learning

Graph Federated Learning (GFL) is a collaborative learning paradigm for graph neural networks that enables multiple agents to jointly train models by sharing model parameters [23, 24]. From the existence of centralized servers, GFL is divided into centralized graph federated learning (C-GFL) and decentralized graph federated learning (D-GFL). C-GFL relies on a server to aggregate model parameters, coordinating cross-client model training [9, 10, 11, 12, 25, 26]. In contrast, D-GFL enables peer-to-peer parameter interaction by constructing communication topology [13, 14, 15, 16].

The server may be impractical in real-world collaborative training, and C-GFL can break down due to server failure or unreliability. The symmetric topology in D-GFL fails to capture agent dependencies. Moreover, model parameters can not obtain sufficient interactive information. Our method solves these problems via graph-driven organizational structure and customized synthetic graph generation.

### 2.2   Graph Lifelong Learning

Graph Lifelong Learning (GLL) is an evolution paradigm that addresses the stable-plasticity dilemma, requiring agents to learn new skills while retaining prior abilities [27, 28, 29]. Existing methods can be categorized into three types: Replay-based methods retain a small number of nodes or subgraphs from old classes, either use generator-produced graphs, to prevent forgetting [19, 21, 22, 30, 31]. Regularization-based approaches introduce regularization terms into the loss function to enhance the preservation of prior knowledge [18, 32, 33, 34, 35]. Parameter isolation methods fully or partially preserve parameters of different tasks via approaches such as dynamic incremental feature extractors, preventing interference between old and new information [20, 36].

The proposed method captures more accurate task prototypes by interacting with synthetic graph information, preventing information forgetting and achieving excellent performance.

## 3   Problem Analysis

### 3.1   Problem Formulation

In heterogeneous dynamic multi-agent graph systems, consider the $A$-agent set, each agent $a$ receives graph data $G_a^t$ at task $t$, which consists of a node set $V_a^t$ and a structure $\alpha_a^t$. A node is associated with node features $X_a^t$ and labels $Y_a^t$. Multiple agents jointly solve a supervised node classification task containing $C^t$ classes at task $t$, so all agents use the same test sets. The graph data $G^t$ is heterogeneous distributed between agents, with overlapping classes, *i.e.*, $\{Y_a^t \cap Y_b^t \neq \emptyset \mid a \neq b\}$. New graph data constantly emerges until time $T$, and each agent needs to learn new knowledge while retaining knowledge from previous tasks. Notably, old tasks are inaccessible and new and old tasks have non-overlapping classes, *i.e.*, $\left\{Y_a^t \cap Y_a^{t'} = \emptyset \mid t \neq t'\right\}$. The relationships between agents can be represented by a directed weighted graph $\mathcal{G}^t = \{\mathcal{A}^t, \mathcal{E}^t, \mathcal{B}^t, M^t\}$, where $\mathcal{A}^t$ is the set of agents, $\mathcal{E}^t$ denotes the edges between agents, $\mathcal{B}^t$ is structure, and $M^t \in \mathbb{R}^{A \times A}$ represents edge weights. For each agent at task $t$, the goal is to collaborate with other agents, share partial information, and learn a graph neural network model that can distinguish new classes from previous classes, expressed as:

$$\theta_a^t = \mathcal{F}_a^t \left( G_a^t, \left\{\mathcal{R}_{a \leftarrow b}^t\right\}_{b=1}^A \mid \mathcal{G}^t \right), \tag{1}$$

where a set of model parameters $\theta_{1:A}^{1:T} = \{\theta_a^t \mid 1 \leq a \leq A, 1 \leq t \leq T\}$ are estimated, $\mathcal{R}_{a \leftarrow b}^t$ is denoted as the information transmitted from agent $b$ to agent $a$. The model is updated through the function $\mathcal{F}_a^t$ based on training data $G_a^t$ and information from external agents.

## 3.2 Graph Socialization Property

**Graph Sociability.** Inspired by prior works [7, 20, 37], we provide graph sociability prediction for agents as follows:

$$p\left(Y_{\text{test},a} \mid G_{\text{test}}, G_{1:A}^{1:T}\right) \approx \underbrace{p\left(Y_{\text{test},a} \mid G_{\text{test}}, \theta_a^{\hat{t}}, \hat{t}\right)}_{\text{Prediction}}, \text{where } \theta_{1:A}^{1:T} = \arg\max_{\theta_{1:A}^{1:T}} \underbrace{p\left(\theta_{1:A}^{1:T} \mid G_{1:A}^{1:T}\right)}_{\text{Parameter Posterior}}, \hat{t} = \arg\max_t \underbrace{p\left(t \mid G_{\text{test}}\right)}_{\text{Task Posterior}}, \quad (2)$$

where $Y_{\text{test},a}$ is prediction results of test graph $G_{\text{test}}$ and $\hat{t}$ denotes optimal task ID. This indicates that effective GSL can be achieved when precise intra-task node prediction, effective collaboration, and accurate task ID query are implemented.

**Parameter Posterior.** Due to complex correlations between agents, we model graph socialized collaboration by maximizing posterior distribution $p\left(\theta_{1:A}^{1:T} \mid G_{1:A}^{1:T}\right)$ of model parameters, rather than simply employing the likelihood function $p\left(G_{1:A}^{1:T} \mid \theta_{1:A}^{1:T}\right)$ [38]. According to Bayes' rule, the posterior distribution can be decomposed as $p\left(\theta_{1:A}^{1:T} \mid G_{1:A}^{1:T}\right) \propto p\left(G_{1:A}^{1:T} \mid \theta_{1:A}^{1:T}\right) p\left(\theta_{1:A}^{1:T}\right)$, where the first probability represents the likelihood of agent model parameters on their respective graph, while the second probability is the prior distribution of parameters, which describes the collaborative patterns between agents.

We assume that the graph arrives independently at each agent and satisfies conditional independence, so we get $p\left(G_{1:A}^{1:T} \mid \theta_{1:A}^{1:T}\right) = \prod_{t=1}^T \prod_{a=1}^A p\left(G_a^t \mid \theta_a^t\right)$, and $-\log p\left(G_a^t \mid \theta_a^t\right)$ corresponds to the loss of agent's model parameters $\theta_a^t$ on graph $G_a^t$. In addition, we capture the pairwise correlations between multiple agent model parameters through collaborative graph optimization [39, 40]. Agents benefit from learning from others who are both complementary and similar, with greater weight given to those who are more complementary and similar. The prior distribution of model parameters can be expressed as $-\log p\left(\theta_a^t\right) \propto$ $\sum_{a=1}^A N_a^t \left(\mathcal{L}_a^t \left(\sum_{b=1}^A M_{ab}^t \theta_b^t; G_a^t\right) + w_{\mathcal{C}} \sum_{b=1}^A \mathcal{C}\left(M_{ab}^t; G_a^t; G_b^t\right) - w_{\mathcal{S}} \sum_{b=1}^A \mathcal{S}\left(M_{ab}^t; \theta_a^t; \theta_b^t\right)\right)$, where the relative size of agents is denoted by $N_a^t = |V_a^t|/N^t$ and $N^t$ is the number of nodes of all agents. $\mathcal{L}_a^t(\cdot)$ represents the empirical risk of its local graph. $\mathcal{C}$ and $\mathcal{S}$ measures complementarity and similarity between agents, as well as $w_{\mathcal{C}}$ and $w_{\mathcal{S}}$ are weights to balance them.

Given the ambiguous nature of an agent's empirical loss, the relative graph size is adopted as a surrogate because graph size is a key measure of credibility. The posterior distribution maximization of parameters $\max p\left(\theta_{1:A}^{1:T} \mid G_{1:A}^{1:T}\right)$ can be rewritten as:

$$\min_{\theta_a^t, M} \sum_{t=1}^T \sum_{a=1}^A \left( -\log p\left(G_a^t \mid \theta_a^t\right) + \sum_{b=1}^A \left( \left(M_{ab}^t - \frac{|V_b^t|}{N^t}\right)^2 + w_{\mathcal{C}} \mathcal{C}\left(M_{ab}^t; G_a^t; G_b^t\right) \right. \right.$$
$$\left. \left. - w_{\mathcal{S}} \mathcal{S}\left(M_{ab}^t; \theta_a^t; \theta_b^t\right) \right) \right) \text{ s.t. } \sum_{b=1}^A M_{ab}^t = 1, \forall a; M_{ab}^t \geq 0, \forall a, b. \quad (3)$$

# 4 Methodology

We present single-agent graph learning to acquire outputs (Section 4.1). The proposed GHG consists of three phases: graph-driven organizational structure (Section 4.2), customized interactive medium (Section 4.3), and prototype-based life cycle (Section 4.4). Moreover, the generalization of graph socialization (Section 4.5) is analyzed to provide insights into socialized collaboration. The overview of the proposed GHG is depicted in Figure 2. The pseudo-code can be found in Appendix C.1.

## 4.1 Single-Agent Graph Learning

As stated in prompt-based graph learning method [20, 41], the graph prompts are represented as $\Phi^t = [\phi_1^t, \ldots, \phi_n^t]^\top \in \mathbb{R}^{n \times d^o}$ for task $t$, where $n$ is the number of token vectors $\phi_i^t$ and $d^o$ is the dimension of features. Node attributes are enhanced through weighted combinations of these tokens. The enhanced graph representations are then passed through a frozen pre-trained GNN model $f_a(\cdot)$ to generate node embeddings. Node classification loss for each agent is defined as:

$$\mathcal{L}_{\text{ce},a}^t = \frac{1}{|V_a^t|} \sum_{i \in V_a^t} \text{CE}\left(F_a^t\left(G_a^t\right)_i, Y_{a,i}^t\right), F_a^t\left(G_a^t\right) = h_a^t\left(f_a\left(\alpha_a^t, X_a^t + \Phi_a^t\right)\right), \quad (4)$$

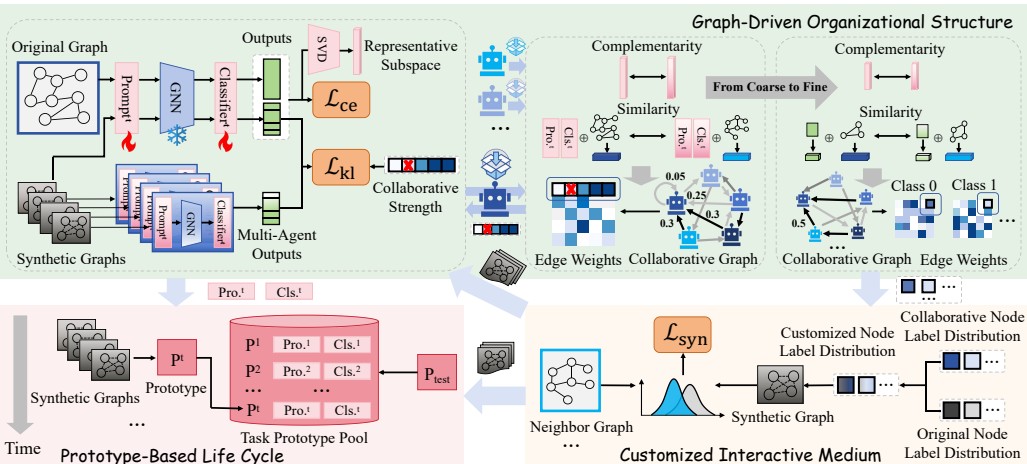

Figure 2: An overview of the GHG method. Firstly, the collaborative graph is built by calculating complementarity and similarity. Next, coarse-grained agent collaboration is used to get collaborative strength for information transfer, while fine-grained customized collaboration aids in initializing label distribution of synthetic graphs. Then, synthetic graphs are generated through distribution alignment. Finally, task prototypes are added gradually, and the task ID is queried to select optimal parameters.

where $V_a^t$ is the training set of $G_a^t$, CE is the cross-entropy loss, $Y_a^t$ is the predicted label, $F_a^t$ denotes node classification function, and $h_a^t$ represents MLP-based classification head. The graph prompts $\{\Phi^1, \ldots, \Phi^T\}$ and classifiers $\{h^1, \ldots, h^T\}$ learn task-specific information while the frozen GNN leverages general cross-task knowledge acquired from the first task.

## 4.2 Graph-Driven Organizational Structure

**Complementarity and Similarity Calculation.** We employ singular value decomposition on agent outputs to indirectly capture the complementarity of the graph, *i.e.* $F_a^t (\cdot) = \mathcal{U}_a^t \Sigma_a^t (\mathcal{V}_a^t)^\top$. The matrix $\mathcal{U}_a^t$ holds the singular vectors of agent outputs, representing directions in the feature space. We treat the first $k$ columns of $\mathcal{U}_a^t$ as the representative subspace. To measure the complementarity between two agents, we apply the average principal angles $\cos \beta_K^t$ between their corresponding subspaces. The complementarity metric is defined as:

$$\mathcal{C}\left(M_{ab}^t; G_a^t; G_b^t\right) = M_{ab}^t \cos\left(\frac{1}{k} \cdot \sum_K \beta_K^t\right), \cos \beta_K^t = \max_{u \in \mathcal{U}_{a,k}^t, v \in \mathcal{U}_{b,k}^t} u^\top v, K = 1, \ldots, k. \quad (5)$$

A large principal angle indicates high complementarity, whereas an angle near $\pi/2$ suggests subspaces are nearly orthogonal, reflecting substantial divergence in their feature spaces. To integrate topological structure and node information, we perform Laplacian smoothing [20] on the graph $G_a^t$ to obtain topology-aware node embeddings:

$$Z_a^{t,(l)} = \left(I - \left(\hat{D}_a^t\right)^{-\frac{1}{2}} \hat{L}_a^t \left(\hat{D}_a^t\right)^{-\frac{1}{2}}\right)^l X_a^t \left(\hat{D}_a^t\right)^{-\frac{1}{2}}, \quad (6)$$

where $l$ is the number of Laplacian smoothing steps, $I$ is an identity matrix, $\hat{L}_a^t = \hat{D}_a^t - \hat{\alpha}_a^t$ denotes graph Laplacian matrix of $\hat{\alpha}_a^t = \alpha_a^t + I$, and $\hat{D}_a^t(\hat{D}_{ii} = \sum_j \hat{\alpha}_{ij})$ is the diagonal degree matrix of $\hat{\alpha}_a^t$. The parameters are enhanced by concatenating model parameters $\theta_a^t \in \{\Phi_a^t, h_a^t\}$ and the mean of topology-aware node embeddings $Z_a^{t,(l)}$ to promote structural information sharing. We utilize cosine distance between the enhanced parameters of agents to quantify similarity, formulated as:

$$\mathcal{S}\left(M_{ab}^t; \Theta_a^t; \Theta_b^t\right) = M_{ab}^t \frac{\Theta_a^t \cdot \Theta_b^t}{\|\Theta_a^t\| \cdot \|\Theta_b^t\|}, \Theta_a^t = \theta_a^t \big\| \text{MEAN}(Z_a^{t,(l)}). \quad (7)$$

**Coarse-Grained Agent Collaboration.** We present edge weights of the collaboration graph between agents derived through coarse-grained collaboration, which are specified as follows:

$$\min_{M_{a*}^t} \sum_{b=1}^{A} \left( \left( M_{ab}^t - \frac{|V_b^t|}{N^t} \right)^2 + w_{\mathcal{C}} \mathcal{C} \left( M_{ab}^t; G_a^t \| \overline{G}_{a*}^t; G_b^t \| \overline{G}_{b*}^t \right) - w_{\mathcal{S}} \mathcal{S} \left( M_{ab}^t; \Theta_a^t; \Theta_b^t \right) \right) \tag{8}$$

$$\text{s.t.} \sum_{b=1}^{A} M_{ab}^t = 1, \forall a; M_{ab}^t \geq 0, \forall a, b,$$

where $\overline{G}_{a*}^t = \|_{b=1}^{\mathcal{N}_a} \widehat{G}_{ab}^t$ represents the synthetic graph (graph synthesis is described in Section 4.3) of neighbor agents $\mathcal{N}_a$ received by agent $a$. We calculate the relative size, complementarity, and similarity of agents based on the number of nodes, subspace representations, and enhanced parameters. A sparse collaborative graph is generated based on edge weights and participation ratio $\delta$, represented as $\mathcal{B}_{ab} = \mathbb{1}\left[ b \in \text{TOPK}(\delta A, M[a, :]) \right]$.

$\overline{G}_{a*}^t$ denotes the joint distribution of the current task, making it an ideal graph for information transfer. In addition to the cross-entropy loss, we incorporate the KL divergence loss to make the agent imitate the predictions of neighboring agents. The overall loss function for agent $a$ at task $t$ is defined as:

$$\mathcal{L}_{\text{all},a}^t = \mathcal{L}_{\text{ce}} \left( F_a^t \left( G_a^t \| \overline{G}_{a*}^t \right), Y_a^t \| \overline{Y}_{a*}^t \right) + w_{\text{kl}} \sum_{b=1}^{\mathcal{N}_a} M_{ab}^t \mathcal{L}_{\text{kl}} \left( F_a^t \left( \widehat{G}_{ab}^t \right), F_b^t \left( \widehat{G}_{ab}^t \right) \right), \tag{9}$$

where $w_{\text{kl}}$ is the loss weight of $\mathcal{L}_{\text{kl}}$. The collaborative graph edge weight $M_{ab}^t$, serving as a measure of collaborative strength, is used as a weight to transfer information from others to a specified agent.

## 4.3 Customized Interactive Medium

**Fine-Grained Customized Collaboration.** To generate sufficient interactive information, we propose fine-grained collaboration, customizing the initial label distribution of synthetic graphs generated by neighboring agents based on their current states. Generally, the relative size of class counts is an important metric. We compute $M_{ab,c}^t$ for each class in the current task by considering relative size, subspace representations, and enhanced embeddings between agents. The enhanced embeddings are concatenated by the mean of outputs and the mean of topology-aware node embeddings. The number of nodes in class $c$ of synthetic graph $\left| \widehat{V}_{ab,c}^t \right|$ generated by agent $b$ for agent $a$ is denoted as follows:

$$\left| \widehat{V}_{ab,c}^t \right| = \frac{\left( \frac{|V_{b,c}^t|}{N_c^t} + w_{\text{col}} M_{ab,c}^t \right)}{N} \cdot \gamma \left| V_b^t \right|, \tag{10}$$

where $N_c^t$ is the sum of the number of nodes of all agents in class $c$, $w_{\text{col}}$ denotes the contribution of fine-grained collaborative graph edge weights, $N$ is the sum of allocation proportions of all classes in agent $b$, and $\gamma$ is the compression ratio.

**Synthetic Graph Generation.** We apply synthetic graphs as a medium for information exchange. Mean and standard deviation statistics for embeddings can effectively capture the distribution of each layer. We concatenate embeddings from 0-layer to $l$-layer and form the concatenated $l$-step embeddings for original graph and synthetic graph: $\mathbf{Z} = \left[ X^t; Z^{(1)}; \ldots; Z^{(l)} \right], \mathbf{R} = \left[ \widehat{X}; R^{(1)}; \ldots; R^{(l)} \right]$. The synthesis loss is expressed as:

$$\mathcal{L}_{\text{syn},ab}^t = \sum_{c=1}^{C^t} \lambda_{ab,c} \left( \left\| \mu_{ab,c}^t - \hat{\mu}_{ab,c}^t \right\|_2^2 + w_\sigma \left\| \sigma_{ab,c}^t - \hat{\sigma}_{ab,c}^t \right\|_2^2 \right), \tag{11}$$

where $\mu_c, \hat{\mu}_c$ are the means of $\mathbf{Z}_c, \mathbf{R}_c$ and $\sigma_c, \hat{\sigma}_c$ are the standard deviations of $\mathbf{Z}_c, \mathbf{R}_c$ for every class. $w_\sigma$ is the standard deviation alignment loss weight. $\lambda_{ab,c} = \left( \left| V_{b,c}^t \right| \right)^2 / \left| \widehat{V}_{ab,c}^t \right|$ assigns different weights to different classes. $\widehat{G}_{ab,c}^t$ and $\widehat{V}_{ab,c}^t$ are the synthetic graph and node set generated by agent $b$ for agent $a$. We use a unit matrix as the predefined adjacency matrix, *i.e.*, $\widehat{\alpha}_{ab}^t = I$, which eliminates the need to train an adjacency matrix generator and avoids encoding synthetic graphs [42, 43].

## 4.4 Prototype-Based Life Cycle

On completing training for each task, each agent stores task-specific prompts and classifiers. Consequently, GHG focuses on forecasting the task ID. We construct task prototypes by averaging node representations from synthetic graphs generated for agent $a$ by neighbor agents:

$$P_a^t = \frac{1}{\left|\overline{V}_{a*}^t\right|} \sum_{i \in \overline{V}_{a*}^t} \left\|_{b=1}^{\mathcal{N}_a} R_{ab,i}^{t,(l)}. \tag{12}$$

All task prototypes can be constructed and added as $\mathbf{P}_a = \{P_a^1, \ldots, P_a^T\}$. We apply the mean of $Z_{\text{test}}^{(l)}$ to calculate test prototype, denoted as $\mathbf{P}_{\text{test},a}$. Subsequently, we query task prototype pool using $\mathbf{P}_{\text{test},a}$ and retrieve task ID $\hat{t}$ that exhibits the highest similarity to $\mathbf{P}_{\text{test},a}$. We leverage the prompts and classifier corresponding to task $\hat{t}$, together with the fixed GNN model, to make predictions.

## 4.5 Graph Socialization Generalization

We analyze the graph sociability influence of minimizing overall loss (9) on the generalization of multi-agent graph distribution for each task ($t$ is omitted for simplicity) in Theorem 1. Proposition 1 indicates that the generalization bound in Theorem 1 is tighter than that of training on a single graph.

**Theorem 1.** *The overall graph distribution of agent $a$ come from three aspects: $\mathcal{D}_a^{all} = \omega \mathcal{D}_a + \omega^{syn} \mathcal{D}_a^{syn} + \omega^{kl} \mathcal{D}_a^{kl}$. $\{\omega, \omega^{syn}, \omega^{kl}\}$ reflects the agent's demand on single graphs, synthetic graphs, and agent predictions from neighbors. Denote the model as $F_a = \Phi_a \circ h_a \in \psi_a \circ \mathcal{H}_a = \mathcal{F}_a$. Inspired by [44, 45, 46, 47], the generalization error on multi-agent graph distribution is bounded as follows:*

$$\epsilon_{\mathcal{D}_A}(F_a) \le \epsilon_{\mathcal{D}_a^{all}}(F_a) + \omega \mathbf{C}(\mathcal{D}_a, \mathcal{D}_A) + \omega^{syn} \epsilon_{\mathcal{D}_A}(\mathcal{Y}_a^{syn}) + \omega^{kl} \epsilon_{\mathcal{D}_A}(\mathcal{Y}_a^{kl}) + \frac{(1-\omega)}{2} \mathbf{d}_{\mathcal{H}_a \Delta \mathcal{H}_a}(\psi \circ \mathcal{D}_a^{syn}, \psi \circ \mathcal{D}_A), \tag{13}$$

*where $\epsilon$ denotes error between model $F$ and optimal labeling $\mathcal{Y}$ under distribution $\mathcal{D}$. $\mathbf{C}(\cdot)$ represents a small distance term, and $\mathbf{d}_{\mathcal{H}_a \Delta \mathcal{H}_a}$ measures the distance between two distributions.*

The first term is the error bound on $\mathcal{D}_a^{all}$. The second term is a constant and related to a single graph. The third term quantifies the discrepancy between real labeling $\mathcal{Y}_A$ and synthetic graph labeling $\mathcal{Y}_a^{syn}$, while the fourth term is derived from the KL-divergence loss in the second term of Eq. (9). The synthetic graph generation makes $\mathcal{D}_a^{syn}$ similar to $\mathcal{D}_A$, so $\mathcal{Y}_a^{syn}$ approximates $\mathcal{Y}_A$. The small $\epsilon_{\mathcal{D}_A}(\mathcal{Y}_a^{syn})$ shows that the model trained on $\mathcal{D}_a^{syn}$ closely learns the multi-agent graph distribution, and the task prototype is more holistic in Eq. (12). The last term $\mathbf{d}_{\mathcal{H}_a \Delta \mathcal{H}_a}$ emphasizes the need for distribution alignment in Eq. (11).

**Proposition 1.** *Given the conditions outlined in Theorem 1, we can derive:*

$$\sup_{F \in \mathcal{F}_a} \min\left\{\mathcal{Q}(\mathcal{D}_a^{syn}), \mathcal{Q}(\mathcal{D}_a^{kl})\right\} \le \inf_{F \in \mathcal{F}_a}(\epsilon_{\mathcal{D}_a}(F) - \epsilon_{\mathcal{D}_A}(F)) + \mathbf{C}(\mathcal{D}_a, \mathcal{D}_A), \tag{14}$$

*where $\mathcal{Q}(\mathcal{D}) = |\epsilon_{\mathcal{D}}(F) - \epsilon_{\mathcal{D}_A}(F)| + \epsilon_{\mathcal{D}_A}(\mathcal{Y}^{\mathcal{D}})$, the usage of $\mathcal{D}_a^{syn}$ and $\mathcal{D}_a^{kl}$ yields a tighter generalization bound than using only single graphs.*

When multi-agent graphs are highly heterogeneous, $\inf_{F \in \mathcal{F}_a}(\epsilon_{\mathcal{D}_a}(F) - \epsilon_{\mathcal{D}_A}(F))$ and $\mathbf{C}(\mathcal{D}_a, \mathcal{D}_A)$ become large. Since $\mathcal{D}_a^{syn}$ and $\mathcal{D}_a^{kl}$ are close to the multi-agent distribution, the left side of Eq. (14) is small. Thus, Proposition 1 indicates that when multi-agent graphs are extremely heterogeneous and $\mathcal{D}_a^{syn}$ and $\mathcal{D}_a^{kl}$ approximate multi-agent graphs, agents depend more on $\mathcal{D}_a^{syn}$ and $\mathcal{D}_a^{kl}$ for better collaboration. Theorem 1 and Proposition 1 are proved in Appendix B.

# 5 Experiments

## 5.1 Datasets and Setups

**Datasets and Settings.** We assess the effectiveness of GHG by leveraging seven publicly available datasets, and their statistical details are presented in Appendix C.2. CoraFull [48], Arxiv [49], and Reddit [50] include 7, 4, and 4 tasks, respectively, each containing 10 classes. Cora [51] and CiteSeer [51] both comprise 3 tasks with 2 classes per task. SLAP [52] and Computers [53] include 3 and 2 tasks, respectively, each with 5 classes. $A$ is set to 5 for all datasets except Cora and CiteSeer, where it is 2. We generate $A$ heterogeneous agents using Dirichlet partitioning [54], where the allocation ratio of nodes for each class is denoted as $\text{Dir}_A(het)$. We define heterogeneity level as $het = 0.1$ (strong) and $het = 0.5$ (weak). Implementation details are provided in Appendix C.4.

Table 1: Performance comparison on five datasets in strong heterogeneity setups. Results are averaged among three trials. The best and second results are highlighted in **bold** and underline.

| Dataset | Paradigm | CoraFull | | Arxiv | | Reddit | | Cora | | CiteSeer | |
|---|---|---|---|---|---|---|---|---|---|---|---|
| Metric | | MAP↑ | MAF↓ | MAP↑ | MAF↓ | MAP↑ | MAF↓ | MAP↑ | MAF↓ | MAP↑ | MAF↓ |
| Single | - | $4.2_{\pm0.1}$ | $36.7_{\pm0.4}$ | $8.4_{\pm0.0}$ | $35.1_{\pm0.2}$ | $15.8_{\pm0.1}$ | $60.1_{\pm0.5}$ | $18.2_{\pm0.1}$ | $54.9_{\pm0.1}$ | $16.9_{\pm0.0}$ | $54.9_{\pm0.0}$ |
| FedAvg | FL | $4.6_{\pm0.1}$ | $38.3_{\pm0.2}$ | $8.9_{\pm0.1}$ | $36.1_{\pm0.3}$ | $26.6_{\pm0.2}$ | $47.9_{\pm0.7}$ | $18.2_{\pm0.1}$ | $58.7_{\pm0.1}$ | $17.0_{\pm0.0}$ | $54.9_{\pm0.1}$ |
| DFedGNN | GFL | $4.6_{\pm0.2}$ | $39.5_{\pm0.4}$ | $9.0_{\pm0.1}$ | $36.0_{\pm0.4}$ | $14.8_{\pm0.1}$ | $62.7_{\pm0.6}$ | $18.0_{\pm0.0}$ | $54.4_{\pm1.8}$ | $17.0_{\pm0.0}$ | $55.4_{\pm0.1}$ |
| Fed-PUB | GFL | $4.0_{\pm0.1}$ | $34.0_{\pm0.3}$ | $7.7_{\pm0.2}$ | $35.1_{\pm0.1}$ | $13.5_{\pm0.3}$ | $48.1_{\pm0.6}$ | $16.9_{\pm0.1}$ | $53.0_{\pm0.2}$ | $16.7_{\pm0.0}$ | $55.9_{\pm0.2}$ |
| FedGTA | GFL | $4.6_{\pm0.1}$ | $38.1_{\pm0.4}$ | $8.7_{\pm0.1}$ | $35.8_{\pm0.4}$ | $14.9_{\pm0.1}$ | $62.4_{\pm0.7}$ | $18.0_{\pm0.0}$ | $59.7_{\pm2.4}$ | $17.0_{\pm0.0}$ | $55.0_{\pm0.1}$ |
| FedTAD | GFL | $4.6_{\pm0.2}$ | $37.9_{\pm0.4}$ | $8.9_{\pm0.1}$ | $36.0_{\pm0.1}$ | $24.4_{\pm0.2}$ | $50.0_{\pm0.4}$ | $18.2_{\pm0.2}$ | $54.8_{\pm0.8}$ | $17.0_{\pm0.0}$ | $54.8_{\pm0.0}$ |
| TWP | GLL | $5.1_{\pm0.2}$ | $34.9_{\pm0.2}$ | $8.4_{\pm0.1}$ | $35.1_{\pm0.1}$ | $16.2_{\pm0.2}$ | $58.6_{\pm0.5}$ | $18.6_{\pm0.1}$ | $55.2_{\pm0.1}$ | $17.0_{\pm0.0}$ | $54.9_{\pm0.1}$ |
| ERGNN | GLL | $23.2_{\pm0.3}$ | $14.0_{\pm0.3}$ | $19.6_{\pm0.6}$ | $18.2_{\pm0.6}$ | $36.5_{\pm0.7}$ | $31.6_{\pm0.9}$ | $24.0_{\pm4.3}$ | $42.5_{\pm5.7}$ | $17.9_{\pm0.1}$ | $53.4_{\pm0.1}$ |
| GSIP | GLL | $26.3_{\pm0.9}$ | $12.8_{\pm0.8}$ | $22.5_{\pm0.9}$ | $9.7_{\pm0.2}$ | $48.4_{\pm0.6}$ | $15.3_{\pm0.4}$ | $33.2_{\pm0.9}$ | $34.5_{\pm2.4}$ | $19.6_{\pm0.5}$ | $50.6_{\pm0.7}$ |
| TPP | GLL | $28.9_{\pm0.2}$ | $4.8_{\pm0.3}$ | $16.4_{\pm0.3}$ | $6.0_{\pm0.2}$ | $43.4_{\pm0.2}$ | $5.4_{\pm0.6}$ | $53.6_{\pm1.6}$ | $\mathbf{0.0_{\pm0.0}}$ | $51.6_{\pm0.1}$ | $\mathbf{0.0_{\pm0.0}}$ |
| DMSG | GLL | $27.3_{\pm0.6}$ | $9.0_{\pm1.0}$ | $21.8_{\pm0.1}$ | $8.8_{\pm0.2}$ | $47.9_{\pm0.2}$ | $13.2_{\pm1.6}$ | $37.5_{\pm3.5}$ | $27.7_{\pm5.3}$ | $33.4_{\pm0.8}$ | $28.7_{\pm1.3}$ |
| Fed-TPP | GFLL | $28.6_{\pm0.3}$ | $4.4_{\pm0.3}$ | $15.3_{\pm0.2}$ | $\underline{5.3_{\pm0.2}}$ | $40.1_{\pm0.1}$ | $5.3_{\pm0.6}$ | $\underline{58.7_{\pm1.7}}$ | $\mathbf{0.0_{\pm0.0}}$ | $\underline{51.7_{\pm0.1}}$ | $\mathbf{0.0_{\pm0.0}}$ |
| Fed-DMSG | GFLL | $\underline{37.3_{\pm0.3}}$ | $\underline{0.1_{\pm0.5}}$ | $\underline{22.7_{\pm0.3}}$ | $7.9_{\pm0.6}$ | $57.4_{\pm0.7}$ | $6.4_{\pm0.4}$ | $39.9_{\pm1.6}$ | $\underline{18.1_{\pm3.2}}$ | $40.1_{\pm1.0}$ | $14.1_{\pm1.3}$ |
| POWER | GFLL | $25.3_{\pm0.4}$ | $14.8_{\pm0.6}$ | $11.0_{\pm0.1}$ | $25.7_{\pm0.3}$ | $\underline{61.4_{\pm1.0}}$ | $2.9_{\pm1.5}$ | $40.9_{\pm0.3}$ | $27.1_{\pm0.7}$ | $36.9_{\pm2.5}$ | $24.0_{\pm3.8}$ |
| MASC | SL | $36.7_{\pm0.4}$ | $0.9_{\pm1.2}$ | $22.3_{\pm0.5}$ | $16.5_{\pm0.7}$ | $58.1_{\pm1.8}$ | $4.6_{\pm3.2}$ | $39.2_{\pm2.3}$ | $23.3_{\pm4.3}$ | $36.6_{\pm0.2}$ | $24.3_{\pm0.5}$ |
| GHG | GSL | $\mathbf{54.0_{\pm0.8}}$ | $\mathbf{0.0_{\pm0.0}}$ | $\mathbf{59.2_{\pm0.3}}$ | $\mathbf{0.0_{\pm0.0}}$ | $\mathbf{86.7_{\pm0.9}}$ | $\mathbf{0.0_{\pm0.0}}$ | $\mathbf{79.2_{\pm3.4}}$ | $\mathbf{0.0_{\pm0.0}}$ | $\mathbf{65.7_{\pm4.1}}$ | $\mathbf{0.0_{\pm0.0}}$ |

**Baselines and Metrics.** We compare GHG with following baselines: Single, five FL/GFL methods (*i.e.* FedAvg [55], DFedGNN [13], Fed-PUB [9], FedGTA [25], and FedTAD [10]), five GLL methods (*i.e.* TWP [18], ERGNN [19], GSIP [21], TPP [20], and DMSG [22]), graph federated lifelong learning methods (*i.e.* FedAvg combines with two representative GLL methods and POWER [56]), and socialized learning method MASC [7]. We evaluate the performance of compared methods using two widely adopted metrics: Average Performance (AP) and Average Forgetting (AF) [34]. The final results are reported as mean values across all agents, denoted as MAP and MAF.

## 5.2 Performance Comparison

**GHG facilitates multi-agent graph socialized learning effectively.** The effects of GSL for $het$=0.1 on seven datasets are presented in Tables 1 and 2, and results for $het$=0.5 are in Appendix D.1. GHG surpasses all baselines in performance. Results are discussed as follows:

(1) **FL/GFL methods** cannot sufficiently transmit information. Consequently, collaborative collapse of heterogeneity is exacerbated in the dynamic environment, resulting in lower performance. D-GFL method DFedGNN relies on an undirected symmetric communication topology, which can not satisfy the collaborative demands of agents. In particular, GFL performs poorly compared with FedAvg. The reason might be that they focus on learning single fixed data and even overfit, thus lacking cooperation. We build an organizational structure based on directed weighted graphs and a customized interactive medium to achieve autonomous collaboration and sufficient information interaction.

(2) **GLL methods** are limited to single agent settings. Moreover, old information is forgotten after learning new tasks, so forgetting still occurs. TPP and Fed-TPP achieve high MAP and zero MAF on Cora and Citeseer datasets. This arises from agents only having two categories, leading to minimal divergence between individual and overall data distributions and enabling accurate task ID prediction. The prototypes formed by GHG, leveraging interactive information in strong heterogeneity, can still predict the task ID with 100% accuracy. Thus, it achieves zero MAF across all datasets, retaining old information losslessly.

(3) **GFLL methods** offer some mitigation for these issues, but a considerable performance disparity remains when compared with GHG. Merely integrating federated and lifelong learning fails to adequately address information collapse and forgetting. Conversely, GHG achieves information sharing and accumulation by designing organizational structure, interactive medium, and life cycle.

(4) **SL method** MASC forgot several old knowledge, and full-data interaction causes high transmission. Our GHG successfully preserves old information and avoids interaction costs by leveraging synthetic graphs as an effective medium.

Table 2: Performance comparison on SLAP and Computers datasets in strong heterogeneity setups.

| Dataset | Para. | SLAP | | Computers | |
|---|---|---|---|---|---|
| Metric | | MAP↑ | MAF↓ | MAP↑ | MAF↓ |
| Single | - | $8.2_{\pm0.1}$ | $34.1_{\pm1.8}$ | $13.6_{\pm2.0}$ | $30.8_{\pm7.9}$ |
| FedAvg | GFL | $8.0_{\pm0.1}$ | $32.9_{\pm0.6}$ | $10.5_{\pm0.4}$ | $4.9_{\pm2.9}$ |
| DFedGNN | GFL | $8.1_{\pm0.1}$ | $33.8_{\pm0.3}$ | $19.8_{\pm2.2}$ | $21.8_{\pm5.3}$ |
| Fed-PUB | GFL | $11.6_{\pm0.1}$ | $42.3_{\pm1.9}$ | $27.5_{\pm1.1}$ | $64.7_{\pm2.9}$ |
| FedGTA | GFL | $8.3_{\pm0.2}$ | $33.7_{\pm2.3}$ | $13.5_{\pm5.2}$ | $10.6_{\pm4.2}$ |
| FedTAD | GFL | $8.1_{\pm0.1}$ | $31.8_{\pm2.2}$ | $10.5_{\pm0.4}$ | $5.0_{\pm6.7}$ |
| TWP | GLL | $8.1_{\pm0.1}$ | $33.2_{\pm1.5}$ | $14.5_{\pm1.0}$ | $29.2_{\pm4.2}$ |
| ERGNN | GLL | $15.6_{\pm0.2}$ | $20.8_{\pm1.1}$ | $18.0_{\pm2.0}$ | $24.1_{\pm6.1}$ |
| GSIP | GLL | $15.7_{\pm0.5}$ | $15.9_{\pm0.7}$ | $19.2_{\pm1.0}$ | $36.4_{\pm5.2}$ |
| TPP | GLL | $24.8_{\pm1.3}$ | $15.8_{\pm2.0}$ | $\underline{36.7_{\pm0.6}}$ | $\mathbf{0.0_{\pm0.0}}$ |
| DMSG | GLL | $15.8_{\pm0.2}$ | $13.6_{\pm0.5}$ | $28.5_{\pm2.4}$ | $30.7_{\pm5.2}$ |
| Fed-TPP | GFLL | $\underline{27.2_{\pm1.2}}$ | $12.3_{\pm0.7}$ | $36.5_{\pm1.5}$ | $\mathbf{0.0_{\pm0.0}}$ |
| Fed-DMSG | GFLL | $18.0_{\pm0.3}$ | $\underline{9.8_{\pm0.7}}$ | $32.2_{\pm3.1}$ | $\underline{3.6_{\pm2.8}}$ |
| POWER | GFLL | $15.5_{\pm0.5}$ | $20.1_{\pm0.2}$ | $10.7_{\pm3.2}$ | $24.4_{\pm7.3}$ |
| MASC | SL | $15.0_{\pm0.3}$ | $18.6_{\pm0.8}$ | $24.2_{\pm1.6}$ | $21.6_{\pm1.3}$ |
| GHG | GSL | $\mathbf{62.4_{\pm3.0}}$ | $\mathbf{0.0_{\pm0.0}}$ | $\mathbf{82.5_{\pm0.2}}$ | $\mathbf{0.0_{\pm0.0}}$ |

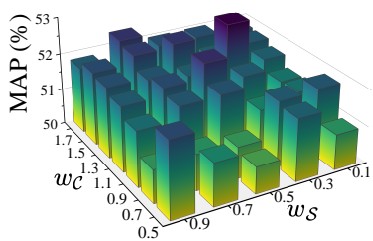

(a) Complementarity weight $w_{\mathcal{C}}$ and similarity weight $w_{\mathcal{S}}$

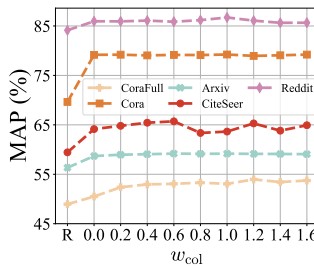

(b) Collaborative weight $w_{\text{col}}$

Figure 3: The analysis of hyper-parameters.

## 5.3 Ablation Study

To explore the contributions of different components, we perform an ablation study in terms of the MAP metric. Table 3 shows progressive improvements with each component, confirming their importance to GSL. We devise four variants for GHG—**B**: This variant uses TPP [20] as baseline; **B+$\mathcal{L}_{\text{syn}}$**: This vari-

Table 3: Ablation comparisons on five datasets.

| Method | CoraFull | Arxiv | Reddit | Cora | CiteSeer |
|---|---|---|---|---|---|
| B | $28.9_{\pm0.2}$ | $16.4_{\pm0.3}$ | $43.4_{\pm0.2}$ | $53.6_{\pm1.6}$ | $51.6_{\pm0.1}$ |
| B+$\mathcal{L}_{\text{syn}}$ | $33.2_{\pm0.3}$ | $26.4_{\pm0.3}$ | $48.3_{\pm0.2}$ | $54.1_{\pm0.2}$ | $52.6_{\pm0.7}$ |
| B+$\mathcal{L}_{\text{syn}}$+$\mathcal{L}_{\text{ce}}$ | $51.6_{\pm1.0}$ | $58.9_{\pm0.3}$ | $83.4_{\pm0.9}$ | $78.8_{\pm4.2}$ | $63.6_{\pm2.2}$ |
| B+$\mathcal{L}_{\text{syn}}$+$\mathcal{L}_{\text{ce}}$+$\mathcal{L}_{\text{kl}}$ | $\mathbf{54.0_{\pm0.8}}$ | $\mathbf{59.2_{\pm0.3}}$ | $\mathbf{86.7_{\pm0.9}}$ | $\mathbf{79.2_{\pm3.4}}$ | $\mathbf{65.7_{\pm4.1}}$ |

ant constructs accurate task prototypes leveraging synthetic graph loss; **B+$\mathcal{L}_{\text{syn}}$+$\mathcal{L}_{\text{ce}}$**: This variant incorporates synthetic graphs from neighbor agents into single graph training with cross-entropy loss; **B+$\mathcal{L}_{\text{syn}}$+$\mathcal{L}_{\text{ce}}$+$\mathcal{L}_{\text{kl}}$**: This variant is the full model, trained with KL divergence loss to help agents mimic others' patterns, achieving the best performance.

## 5.4 Further Analysis

**Hyper-Parameter Analysis.** (1) We assess the performance on CoraFull under diverse combinations of trade-off parameters. Specifically, we tune $w_{\mathcal{C}}$ varies in 0.5 to 1.7 and $w_{\mathcal{S}}$ varies in 0.1 to 0.9. As shown in Figure 3(a), the performance of GHG is stable across $w_{\mathcal{C}}$ and $w_{\mathcal{S}}$ varies, indicating low sensitivity to hyper-parameters. (2) Figure 3(b) presents performance as $w_{\text{col}}$ varies, where "R" indicates random sampling. The customized graph synthesis significantly outperforms the random approach, with an almost 10% improvement on Cora, demonstrating the effectiveness of the customized interactive medium. The analysis of loss weights is presented in Appendix D.2.

**Graph Socialization Necessity.** The performance of GHG with 5, 10, and 20 agents, as well as different participation ratios $\delta$ is shown in Table 4(a). We can observe that the higher the participation ratio, the higher the performance, which proves the necessity of agent socialization. The more agents $A$, the more dispersed the graph becomes, thus resulting in lower performance. However, the only special case is when $\delta$ equals 0.2. In the case of 5 or 10 agents, only 1 or 2 agents cooperate, leading to the inability to predict task ID accurately, so performance is lower than that of 20 agents.

**Graph Socialization Efficiency.** (1) We compare GHG with baselines on Corafull across different interaction rounds in Figure 4(b). GHG outperforms baselines in all cases. It peaks at four interactions, which shows GHG efficiently completes collaboration. In contrast, Fed-DMSG and MASC require

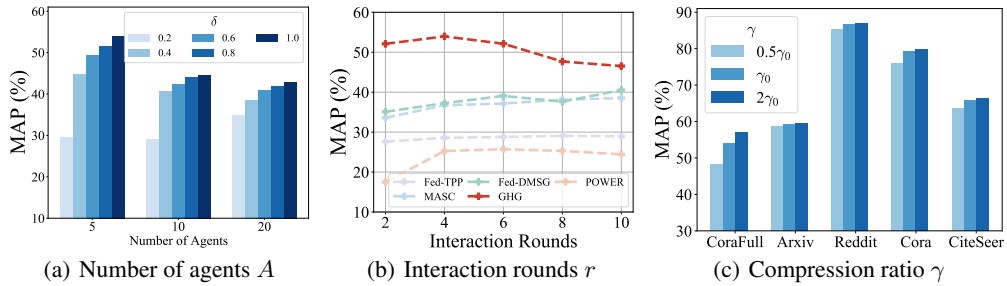

| (a) Number of agents $A$ | (b) Interaction rounds $r$ | (c) Compression ratio $\gamma$ |

Figure 4: The analysis of parameters.

more interactions for high performance. Fed-TPP shows little variation with different $r$, and POWER fails to save interaction overhead. (2) Figure 4(c) illustrates the performance variation on five datasets with different compression ratios. We observe that higher $\gamma$ generally leads to better performance. However, we use the results with $\gamma_0$ to limit interaction costs. Even when $\gamma$ is halved, our method maintains high performance. We further analyze the time, memory, and communication costs in Appendix D.3 to demonstrate the efficiency of GHG over baselines.

**Visulization.** (1) The visualization of collaborative graph edge weights is displayed in Figure 5. The first row shows three matrices of coarse-grained agent collaboration for Task 0 of Reddit, with those for the other three tasks in Appendix D.4. The second row presents three matrices of fine-grained customized collaboration for Class 0 of Task 0. Agents almost abandon their initial state in the complementarity matrix, impeding the subsequent training. In the similarity matrix, the insistence of agents on their models hinders information acquisition. Our collaborative approach filters out redundant interaction information and gets what each agent needs, highlighting the role of GHG in enhancing autonomous socialization efficiency. (2) As shown in Figure 6, we utilize t-SNE [57] to visualize node embeddings of three classes generated by random sampling and GHG on Arxiv. The reciprocal generation of node representations by two agents within GHG facilitates preservation of the category-wise node distributions inherent in the original graph and enables the customization of category quantities to meet agent demands. This demonstrates that GHG has a superior ability to capture sufficient graph information.

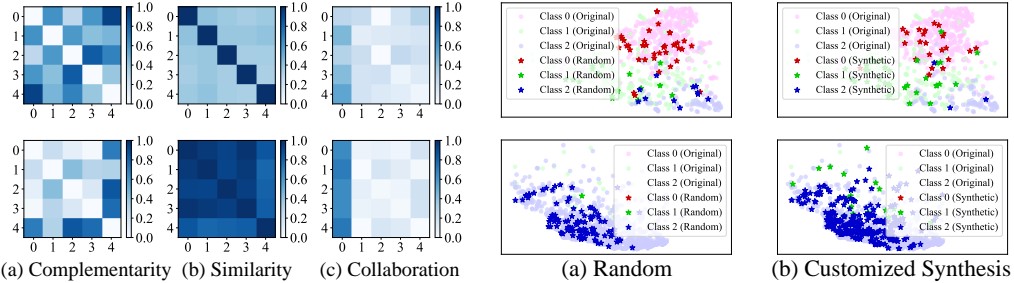

| (a) Complementarity (b) Similarity (c) Collaboration | (a) Random (b) Customized Synthesis |

Figure 5: The visualization of collaborative graph edge weight matrices on Reddit dataset.

Figure 6: The visualization of node embeddings on Arxiv dataset.

## 6 Conclusion

Current paradigms face the predicament of information collapse and forgetting, struggling to share and accumulate information in dynamic heterogeneous graph settings collaboratively. We present a practical GSL paradigm and develop a GHG approach to enhance the performance of each agent. Graph-driven organizational structure, customized interactive medium, and prototype-based life cycle form three key elements of GSL. In future work, we aim to refine the organizational structure further to accommodate large-scale multi-agent systems and strengthen the scalability of the graph socialized learning model. Moreover, we plan to evaluate the effectiveness of the graph socialized learning method on real-world application datasets.

## Acknowledgement

This work was supported in part by the National Science and Technology Major Project under Grant 2022ZD0116500, in part by the National Natural Science Foundation of China under Grants 62436002, 62476195, U23B2049, and 62222608, in part by Tianjin Natural Science Funds for Distinguished Young Scholar under Grant 23JCJQJC00270, in part by Zhejiang Provincial Natural Science Foundation of China under Grant LD24F020004, in part by Tianjin Young Scientific and Technological Talents Project under grant QN20230305, in part by Tianjin Science and Technology Plan Project under grant 24YDTPJC00150, and in part by Natural Science Foundation of Tianjin under grant 24JCYBJC00950.

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

# A  Appendix

The appendix contains supplementary proofs, together with comprehensive details of implementation and results mentioned in the main paper. It is organized as follows:

- In Section B, we provide the proofs of Theorem 1 and Proposition 1 regarding graph socialization generalization.

- In Section C, it presents a thorough overview of baselines, detailed descriptions of datasets, and relevant experimental details in the main paper.

- In Section D, it offers complete experimental results alongside in-depth analysis to fully evaluate the performance of the proposed method, followed by time, memory, and communication costs analysis.

- In Section E, we discuss the related work, privacy issue, limitation, and potential broader impact of the method.

# B  Proof of Graph Socialization Generalization

## B.1  Proof of Theorem 1

*Proof.* We leverage previous results [44, 46, 47] to get the inequality and multi-agent graph distribution generalization bound:

$$\epsilon_{\mathcal{D}_A}(F_a) \leq \epsilon_{\mathcal{D}_a}(F_a) + \mathbf{C}(\mathcal{D}_a, \mathcal{D}_A), \tag{15}$$

$$\epsilon_{\mathcal{D}_A}(F) \leq \epsilon_{\mathcal{D}}(F) + \frac{1}{2}\mathbf{d}_{\mathcal{H}\Delta\mathcal{H}}(\psi \circ \mathcal{D}, \psi \circ \mathcal{D}_A) + \epsilon_{\mathcal{D}_A}(\mathcal{Y}), \tag{16}$$

where $\mathbf{C}(\mathcal{D}_a, \mathcal{D}_A) = \frac{1}{2}\mathbf{d}_{\mathcal{F}_a\Delta\mathcal{F}_a}(\mathcal{D}_a, \mathcal{D}_A) + \eta(\mathcal{D}_a)$, and $\eta(\mathcal{D}_a) = \min_{F \in \mathcal{F}_a} \epsilon_{\mathcal{D}_a}(F) + \epsilon_{\mathcal{D}_A}(F)$ is a constant. These terms tend to be small except when graph heterogeneity is severe [44]. $\psi_a, \mathcal{H}_a, \mathcal{F}_a$ are the space of prompts, classifier, and model on agent $a$. $\mathcal{Y}$ represents the optimal labeling function of graph distribution $\mathcal{D}$. Since the overall graph distribution of agent $a$ originates from single graphs, synthetic graphs, and predictions of neighboring agents on synthetic graphs, we sum three inequalities after multiplying them by their corresponding weights.

The support of $\mathcal{D}_a^{\mathrm{syn}}$ and $\mathcal{D}_a^{\mathrm{kl}}$ are identical. As a consequence, their proximity to the support of the multi-agent graph distribution also matches, which is denoted as:

$$\mathbf{d}_{\mathcal{H}_a\Delta\mathcal{H}_a}(\psi \circ \mathcal{D}_a^{\mathrm{syn}}, \psi \circ \mathcal{D}_A) = \mathbf{d}_{\mathcal{H}_a\Delta\mathcal{H}_a}(\psi \circ \mathcal{D}_a^{\mathrm{kl}}, \psi \circ \mathcal{D}_A). \tag{17}$$

By leveraging the findings from [45] and Pinsker inequality, we establish that the optimization of loss (9) leads to the minimization of $\epsilon_{\mathcal{D}_a^{\mathrm{all}}}(F_a)$. After gradually adding three inequalities, we obtain:

$$
\begin{aligned}
\epsilon_{\mathcal{D}_A}(F_a) \leq & \epsilon_{\mathcal{D}_a^{\mathrm{all}}}(F_a) + \mathbf{C}(\mathcal{D}_a, \mathcal{D}_A) + \frac{\omega^{\mathrm{syn}}}{2}\mathbf{d}_{\mathcal{H}_a\Delta\mathcal{H}_a}(\psi \circ \mathcal{D}_a^{\mathrm{syn}}, \psi \circ \mathcal{D}_A) \\
& + \epsilon_{\mathcal{D}_A}(\mathcal{Y}_a^{\mathrm{syn}}) + \frac{\omega^{\mathrm{kl}}}{2}\mathbf{d}_{\mathcal{H}_a\Delta\mathcal{H}_a}(\psi \circ \mathcal{D}_a^{\mathrm{syn}}, \psi \circ \mathcal{D}_A) + \epsilon_{\mathcal{D}_A}(\mathcal{Y}_a^{\mathrm{syn}}) \\
\leq & \epsilon_{\mathcal{D}_a^{\mathrm{all}}}(F_a) + \omega\mathbf{C}(\mathcal{D}_a, \mathcal{D}_A) + \omega^{\mathrm{syn}}\epsilon_{\mathcal{D}_A}(\mathcal{Y}_a^{\mathrm{syn}}) + \omega^{\mathrm{kl}}\epsilon_{\mathcal{D}_A}(\mathcal{Y}_a^{\mathrm{syn}}) \\
& + \frac{(1-\omega)}{2}\mathbf{d}_{\mathcal{H}_a\Delta\mathcal{H}_a}(\psi \circ \mathcal{D}_a^{\mathrm{syn}}, \psi \circ \mathcal{D}_A).
\end{aligned}
\tag{18}
$$

Since $\omega + \omega^{\mathrm{syn}} + \omega^{\mathrm{kl}} = 1$, we can derive:

$$
\begin{aligned}
\epsilon_{\mathcal{D}_A}(F_a) \leq & \epsilon_{\mathcal{D}_a^{\mathrm{all}}}(F_a) + \omega\mathbf{C}(\mathcal{D}_a, \mathcal{D}_A) + \omega^{\mathrm{syn}}\epsilon_{\mathcal{D}_A}(\mathcal{Y}_a^{\mathrm{syn}}) + \omega^{\mathrm{kl}}\epsilon_{\mathcal{D}_A}(\mathcal{Y}_a^{\mathrm{kl}}) \\
& + \frac{(1-\omega)}{2}\mathbf{d}_{\mathcal{H}_a\Delta\mathcal{H}_a}(\psi \circ \mathcal{D}_a^{\mathrm{syn}}, \psi \circ \mathcal{D}_A).
\end{aligned}
\tag{19}
$$

$\square$

## B.2 Proof of Proposition 1

*Proof.* We initiate with:

$$\sup_{F \in \mathcal{F}_a} \left| \epsilon_{\mathcal{D}_a^{\mathrm{syn}}}(F) - \epsilon_{\mathcal{D}_A}(F) \right| + \epsilon_{\mathcal{D}_A}\left(\mathcal{Y}_a^{\mathrm{syn}}\right) \leq$$

$$\inf_{F \in \mathcal{F}_a} \left( \epsilon_{\mathcal{D}_a}(F) - \epsilon_{\mathcal{D}_A}(F) \right) + \frac{1}{2} \mathbf{d}_{\mathcal{F}_a \Delta \mathcal{F}_a}\left(\mathcal{D}_a, \mathcal{D}_A\right) + \eta\left(\mathcal{D}_a\right), \tag{20}$$

where $\frac{1}{2}\mathbf{d}_{\mathcal{F}_a \Delta \mathcal{F}_a}\left(\mathcal{D}_a, \mathcal{D}_A\right) + \eta\left(\mathcal{D}_a\right) = \mathbf{C}\left(\mathcal{D}_a, \mathcal{D}_A\right)$. For any $F \in \mathcal{F}_a$, we can derive:

$$\epsilon_{\mathcal{D}_a^{\mathrm{syn}}}(F) - \epsilon_{\mathcal{D}_A}(F) + \epsilon_{\mathcal{D}_A}\left(\mathcal{Y}_a^{\mathrm{syn}}\right) \leq \epsilon_{\mathcal{D}_a}(F) - \epsilon_{\mathcal{D}_A}(F) + \mathbf{C}\left(\mathcal{D}_a, \mathcal{D}_A\right)$$

$$\Rightarrow \epsilon_{\mathcal{D}_a^{\mathrm{syn}}}(F) + \epsilon_{\mathcal{D}_A}\left(\mathcal{Y}_a^{\mathrm{syn}}\right) \leq \epsilon_{\mathcal{D}_a}(F) + \mathbf{C}\left(\mathcal{D}_a, \mathcal{D}_A\right). \tag{21}$$

The right side of Eq.(21) corresponds to the original bound stated in [44], so we have:

$$\epsilon_{\mathcal{D}_a^{\mathrm{kl}}}(F) + \epsilon_{\mathcal{D}_A}\left(\mathcal{Y}_a^{\mathrm{kl}}\right) \leq \epsilon_{\mathcal{D}_a}(F) + \frac{1}{2}\mathbf{d}_{\mathcal{F}_a \Delta \mathcal{F}_a}\left(\mathcal{D}_a, \mathcal{D}_A\right) + \eta\left(\mathcal{D}_a\right). \tag{22}$$

Combine Eq.(21) and (22), and if weight $\omega \to 0$,

$$\omega^{\mathrm{kl}}\epsilon_{\mathcal{D}_a^{\mathrm{kl}}}(F) + \omega^{\mathrm{syn}}\epsilon_{\mathcal{D}_a^{\mathrm{syn}}}(F) + \epsilon_{\mathcal{D}_A}\left(\mathcal{Y}_a^{\mathrm{kl}}\right) \leq \epsilon_{\mathcal{D}_a}(F) + \frac{1}{2}\mathbf{d}_{\mathcal{F}_a \Delta \mathcal{F}_a}\left(\mathcal{D}_a, \mathcal{D}_A\right) + \eta\left(\mathcal{D}_a\right). \tag{23}$$

Thus, we determine that our overall graph generalization bound in Theorem 1 is tighter than the single graph generalization bound in [44], provided that the condition in Proposition 1 is satisfied. $\square$

---

**Algorithm 1** Training of GHG

---

**Input:** A series of multi-agent graphs: $G_{1:A}^{1:T} = \{G_a^t \mid 1 \leq a \leq A, 1 \leq t \leq T\}$, task number $T$, interaction round $r$, agent number $A$, the graph neural networks $f_{1:A}(\cdot) = \{f_a(\cdot) \mid 1 \leq a \leq A\}$, initial prompts $\Phi_{1:A}^{1:T} = \{\Phi_a^t \mid 1 \leq a \leq A, 1 \leq t \leq T\}$ and classifiers $h_{1:A}^{1:T} = \{h_a^t \mid 1 \leq a \leq A, 1 \leq t \leq T\}$, initial collaborative graph edge weights $M$.

1: **for** $a = 1$ to $A$ **do**
2:      Pre-train $f_a(\cdot)$ on $G_a^1$ with graph pre-training learning.
3: **end for**
4: **for** $t = 1$ to $T$ **do**
5:      **for** $rnd = 1$ to $r$ **do**
6:          **for** $a = 1$ to $A$ **do**
7:              **if** $rnd == 1$ **then**
8:                  Optimize $\Phi_a^t$ and $h_a^t$ by minimizing single-agent graph learning loss as Eq. (4).
9:              **else**
10:                  Receive interaction information $\mathcal{R} = \{\mathcal{U}_k^t, \mathcal{U}_{c,k}^t, \Phi^t, h^t, \mathrm{MEAN}(F(G^t \| \overline{G}^t)_c),$
        $\mathrm{MEAN}(Z^{t,(l)}), \mathrm{MEAN}(Z_c^{t,(l)}), \widehat{G}^t, F(\widehat{G}^t)\}$ sent from other agents.
11:                  Optimize $\Phi_a^t$ and $h_a^t$ by minimizing multi-agent graph learning loss as Eq. (9).
12:              **end if**
13:              Calculate complementarity $\mathcal{C}$ and similarity $\mathcal{S}$ as Eq. (5) and Eq. (7).
14:              Update collaboration graph edge weights $M_{a*}^t$ as Eq. (8).
15:              Calculate initial label distribution of synthetic graphs for neighbor agents as Eq. (10).
16:              Generate synthetic graphs for neighbor agents as Eq. (11).
17:              Send interaction information $\mathcal{R}$ to other agents.
18:              **if** $rnd == r$ **then**
19:                  Obtain task prototype $P_a^t$ of task $t$ as Eq. (12).
20:              **end if**
21:          **end for**
22:      **end for**
23: **end for**

---

---

**Algorithm 2** Inference in GHG

---

**Input:** Graph neural networks $f_{1:A}(\cdot) = \{f_a(\cdot) \mid 1 \le a \le A\}$, graph prompts $\Phi_{1:A}^{1:T} = \{\Phi_a^t \mid 1 \le a \le A, 1 \le t \le T\}$, classifiers $h_{1:A}^{1:T} = \{h_a^t \mid 1 \le a \le A, 1 \le t \le T\}$, task prototypes $\mathbf{P}_{1:A} = \{\mathbf{P}_a \mid 1 \le a \le A\}$, and test graph $G_{\text{test}}$.
**Output:** Prediction results of agents for test graph.
1: **for** $a = 1$ to $A$ **do**
2:      Obtain task prototype $\mathbf{P}_{\text{test},a}$.
3:      Infer the task ID of test graph $\hat{t}$ by querying $\mathbf{P}_a$ with $\mathbf{P}_{\text{test},a}$.
4:      Obtain predictions through $f_a(\cdot)$ and the corresponding graph prompts $\Phi_a^{\hat{t}}$ and classifier $h_a^{\hat{t}}$.
5: **end for**
6: Return Prediction results of agents for the test graph.

---

## C  Implementation Details

### C.1  Algorithm

The proposed method is summarized in Algorithm 1 for training and Algorithm 2 for inference.

### C.2  Datasets and Baselines

The statistical properties of seven datasets are shown in Table 4. On CoraFull dataset, the node label distribution of five agents under two heterogeneous levels for each task is shown in Figure 7. It can be seen that different agents have different classes and numbers of nodes in each task. We introduce the baselines from the main paper as follows:

- **Single** serves as the lower-bound baseline, exclusively utilizing the most recent single graph to update the model.

- Federated Averaging (**FedAvg**) [55] serves as the foundational federated learning method, where the server aggregates the model parameters received from clients and distributes them back to clients.

- Decentralized Federated Graph Neural Network (**DFedGNN**) [13] adopts a decentralized parallel stochastic gradient descent algorithm for training the graph neural network model across a peer-to-peer network topology.

- Federated Personalized Subgraph Learning (**Fed-PUB**) [9] performs weighted averaging during server-side aggregation and learn personalized sparse masks to update the subgraph-relevant subset of the aggregation parameters.

- Federated Graph Topology-aware Aggregation (**FedGTA**) [25] aggregates through topology-aware local smoothing confidence and mixed neighbor features.

- Federated Topology-aware Data-free Knowledge Distillation (**FedTAD**) [10] measures class-wise knowledge reliability at clients and uses a generator to produce pseudo-graphs, transferring reliable knowledge from clients to the global model.

- Topology-aware Weight Preserving (**TWP**) [18] assesses parameter contribution to task performance and topology, and introduces regularization terms to preserve critical parameters.

- Experience Replay Graph Neural Network (**ERGNN**) [19] achieves multiple node selection strategies to extract representative nodes for memory replay.

- Graph Spatial Information Preservation (**GSIP**) [21] prevents catastrophic forgetting by preserving graph spatial information based on low- and high-frequency components.

- Task Profiling and Prompting (**TPP**) [20] enables replay-free and forget-free learning through discriminative graph prompts and task ID predictions.

- Diversified Memory Selection and Generation (**DMSG)** [22] employs buffer selection strategies considering intra-class and inter-class diversity, along with diversified memory generation for replay.

Table 4: Statistics of datasets.

| Datasets | CoraFull | Arxiv | Reddit | Cora | CiteSeer | SLAP | Computers |
|---|---|---|---|---|---|---|---|
| # nodes | 19,793 | 169,343 | 227,853 | 2,708 | 3,327 | 20,419 | 13,752 |
| # edges | 130,622 | 1,166,243 | 114,615,892 | 5,429 | 4,732 | 172,248 | 245,778 |
| # class | 70 | 40 | 40 | 7 | 6 | 15 | 10 |
| # agent | 5 | 5 | 5 | 2 | 2 | 5 | 5 |
| # task | 7 | 4 | 4 | 3 | 3 | 3 | 2 |
| # novel class | 10 | 10 | 10 | 2 | 2 | 5 | 5 |

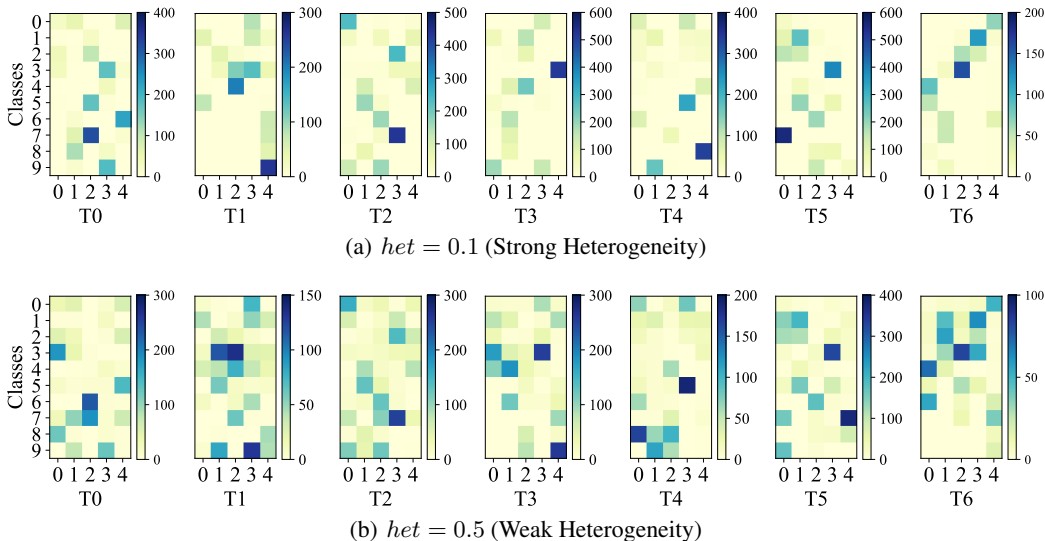

(a) $het = 0.1$ (Strong Heterogeneity)

(b) $het = 0.5$ (Weak Heterogeneity)

Figure 7: Node label distribution for each task of five agents on CoraFull dataset under two heterogeneous levels.

- Graph Evolution Trajectory-aware Knowledge Transfer (**POWER**) [56] replays experience nodes with maximum local-global coverage at clients and tackles global expertise conflict by trajectory-aware knowledge transfer.
- Multi-Agent Socialized Collaboration (**MASC**) [7] constructs collective collaboration and reciprocal altruism and achieves socialized learning via knowledge distillation.

## C.3 Metrics

We use two widely adopted metrics to evaluate the performance of the compared methods: Average Performance (AP) and Average Forgetting (AF) [34]. Specifically, AP and AF are calculated from the lower triangular performance matrix $J \in \mathbb{R}^{T \times T}$, $J^{tt'}(t > t')$ represents the node classification accuracy on task $t'$ after learning task $t$. After learning a new task, average performance assesses the average performance on previous tasks, while average forgetting measures the average performance drop on old tasks. MAP and MAF denote the mean accuracy and forgetting rate across $A$ agents, respectively. Higher MAP and lower MAF correspond to superior performance. MAP and MAF are derived through the following formulas after completing $T$ tasks:

$$\text{MAP} = \frac{1}{A} \sum_{a=1}^{A} \frac{1}{T} \sum_{t=1}^{T} J^{T,t}, \text{MAF} = -\frac{1}{A} \sum_{a=1}^{A} \frac{1}{T-1} \sum_{t=1}^{T-1} \left( J^{T,t} - J^{t,t} \right). \tag{24}$$

## C.4 Detailed Settings

Our model is deployed in PyTorch and on an NVIDIA RTX 3090 GPU. We use Adam with weight decay, setting the learning rate to 0.005 and training for 50 epochs. For graph synthesis of each agent,

Table 5: Performance comparison on five datasets in weak heterogeneity setups. Results are averaged among three trials. The best and second results are highlighted in **bold** and underline.

| Dataset | Paradigm | CoraFull | | Arxiv | | Reddit | | Cora | | CiteSeer | |
|---|---|---|---|---|---|---|---|---|---|---|---|
| Metric | | MAP↑ | MAF↓ | MAP↑ | MAF↓ | MAP↑ | MAF↓ | MAP↑ | MAF↓ | MAP↑ | MAF↓ |
| Single | - | $6.2_{\pm0.1}$ | $55.3_{\pm0.1}$ | $11.9_{\pm0.1}$ | $49.6_{\pm0.1}$ | $21.3_{\pm0.2}$ | $82.1_{\pm0.4}$ | $26.8_{\pm0.1}$ | $79.3_{\pm0.2}$ | $20.4_{\pm0.2}$ | $57.5_{\pm0.5}$ |
| FedAvg | FL | $7.2_{\pm0.3}$ | $59.3_{\pm0.2}$ | $12.3_{\pm0.2}$ | $50.7_{\pm0.4}$ | $21.6_{\pm0.1}$ | $83.6_{\pm0.6}$ | $29.6_{\pm0.5}$ | $81.0_{\pm0.4}$ | $20.4_{\pm0.2}$ | $60.0_{\pm0.2}$ |
| DFedGNN | GFL | $7.3_{\pm0.1}$ | $60.0_{\pm0.3}$ | $12.8_{\pm0.0}$ | $53.0_{\pm0.4}$ | $21.7_{\pm0.0}$ | $83.4_{\pm0.4}$ | $29.0_{\pm0.8}$ | $81.8_{\pm1.0}$ | $23.6_{\pm0.1}$ | $64.1_{\pm0.6}$ |
| Fed-PUB | GFL | $5.3_{\pm0.6}$ | $51.5_{\pm2.2}$ | $11.1_{\pm0.5}$ | $50.7_{\pm0.2}$ | $21.2_{\pm0.2}$ | $84.3_{\pm0.2}$ | $16.7_{\pm0.3}$ | $63.0_{\pm0.5}$ | $16.7_{\pm0.0}$ | $51.3_{\pm1.1}$ |
| FedGTA | GFL | $7.1_{\pm0.2}$ | $59.6_{\pm0.4}$ | $12.6_{\pm0.1}$ | $51.7_{\pm0.1}$ | $21.5_{\pm0.1}$ | $83.3_{\pm0.1}$ | $29.8_{\pm0.5}$ | $81.3_{\pm0.5}$ | $23.6_{\pm0.1}$ | $61.0_{\pm0.0}$ |
| FedTAD | GFL | $7.0_{\pm0.1}$ | $58.5_{\pm0.4}$ | $12.5_{\pm0.1}$ | $52.1_{\pm0.2}$ | $22.5_{\pm0.9}$ | $82.5_{\pm1.3}$ | $31.5_{\pm1.0}$ | $80.3_{\pm0.3}$ | $20.1_{\pm0.1}$ | $59.6_{\pm0.4}$ |
| TWP | GLL | $6.7_{\pm0.2}$ | $51.9_{\pm0.1}$ | $11.9_{\pm0.2}$ | $50.1_{\pm0.3}$ | $21.3_{\pm0.3}$ | $81.8_{\pm0.5}$ | $31.3_{\pm0.1}$ | $79.8_{\pm0.5}$ | $19.9_{\pm0.1}$ | $56.6_{\pm0.3}$ |
| ERGNN | GLL | $37.4_{\pm0.4}$ | $17.7_{\pm0.6}$ | $24.7_{\pm0.5}$ | $29.1_{\pm0.5}$ | $55.6_{\pm0.4}$ | $36.3_{\pm0.4}$ | $36.3_{\pm3.8}$ | $70.6_{\pm5.7}$ | $23.2_{\pm0.2}$ | $56.1_{\pm1.2}$ |
| GSIP | GLL | $43.5_{\pm0.3}$ | $12.5_{\pm0.4}$ | $30.6_{\pm0.2}$ | $14.9_{\pm0.4}$ | $69.4_{\pm1.3}$ | $17.9_{\pm1.7}$ | $44.0_{\pm2.1}$ | $60.3_{\pm3.6}$ | $24.6_{\pm0.7}$ | $53.8_{\pm2.1}$ |
| TPP | GLL | $46.8_{\pm0.4}$ | $1.8_{\pm0.1}$ | $21.0_{\pm0.1}$ | $9.3_{\pm0.2}$ | $69.9_{\pm0.2}$ | $5.1_{\pm0.5}$ | $79.5_{\pm0.4}$ | **$0.0_{\pm0.0}$** | $53.7_{\pm1.3}$ | **$0.0_{\pm0.0}$** |
| DMSG | GLL | $41.4_{\pm0.8}$ | $10.8_{\pm0.9}$ | $25.3_{\pm0.3}$ | $12.9_{\pm0.4}$ | $66.3_{\pm1.1}$ | $16.2_{\pm1.4}$ | $58.3_{\pm2.6}$ | $37.3_{\pm3.5}$ | $41.4_{\pm2.2}$ | $32.8_{\pm2.3}$ |
| Fed-TPP | GFLL | $47.1_{\pm0.1}$ | $2.2_{\pm0.1}$ | $19.5_{\pm0.0}$ | $8.2_{\pm0.2}$ | $67.7_{\pm0.5}$ | $5.0_{\pm0.3}$ | $76.6_{\pm1.7}$ | **$0.0_{\pm0.0}$** | $50.4_{\pm0.5}$ | **$0.0_{\pm0.0}$** |
| Fed-DMSG | GFLL | $56.1_{\pm0.7}$ | $0.4_{\pm0.9}$ | $29.9_{\pm0.4}$ | $8.1_{\pm0.4}$ | $77.0_{\pm1.0}$ | $4.3_{\pm0.2}$ | $67.9_{\pm1.6}$ | $17.1_{\pm1.9}$ | $44.8_{\pm1.2}$ | $27.6_{\pm1.5}$ |
| POWER | GFLL | $42.2_{\pm0.2}$ | $15.8_{\pm0.7}$ | $25.1_{\pm0.8}$ | $32.6_{\pm1.1}$ | $76.0_{\pm0.8}$ | $10.6_{\pm0.6}$ | $63.4_{\pm4.9}$ | $30.1_{\pm4.9}$ | $49.9_{\pm0.7}$ | $21.3_{\pm0.9}$ |
| MASC | SL | $54.7_{\pm0.8}$ | $2.7_{\pm1.1}$ | $31.7_{\pm0.6}$ | $24.9_{\pm0.7}$ | $78.1_{\pm0.4}$ | $7.7_{\pm1.2}$ | $65.1_{\pm1.2}$ | $26.6_{\pm1.5}$ | $48.5_{\pm1.3}$ | $10.6_{\pm1.8}$ |
| GHG | GSL | **$60.7_{\pm0.2}$** | **$0.0_{\pm0.0}$** | **$61.0_{\pm0.2}$** | **$0.0_{\pm0.0}$** | **$92.7_{\pm0.0}$** | **$0.0_{\pm0.0}$** | **$93.9_{\pm0.9}$** | **$0.0_{\pm0.0}$** | **$72.7_{\pm3.7}$** | **$0.0_{\pm0.0}$** |

we employ Adam with a learning rate of 0.001 or 0.005 and 50 epochs per interaction round. The graph prompt settings follow those of TPP [20], with 3 Laplacian steps and three prompts. Results are reported as the mean and standard deviation of three trials. For dataset splits, CoraFull, Arxiv, and Reddit have 60% training, 20% validation, and 20% testing, while Cora, CiteSeer, SLAP, and Computers have 20% training, 40% validation, and 40% testing. The compression ratio $\gamma_0$ is 0.1 for most datasets, except for Reddit, which has a value of 0.01. When applying SVD to agent outputs, the representative subspace column number $k$ of CoraFull, Arxiv, and Reddit is 3, and the last four datasets use $k = 2$. For SVD on outputs of classes, $k$ is set to 1. The number of interactions for the first three datasets is 4, and for the last four datasets, interaction round $r$ is set to 2.

# D  Experimental Results

## D.1  Performance Comparison

The performance under weak heterogeneity conditions is presented in Table 5. Overall, it is better than that under the strong heterogeneity condition. Nevertheless, our method still shows remarkable performance. This indicates the effectiveness of our method in collaboration.

## D.2  Hyper-Parameter Analysis

The KL divergence loss weight $w_{kl}$ takes values in $[1e-3, 1e-2, 1e-1, 1e-0]$, and the results on five datasets are shown in Figure 8(a). Different datasets have different optimal values. We note that the model's performance doesn't vary significantly with different values. The synthetic standard deviation loss weight $w_\sigma$ is set to $[1e-4, 1e-3, 1e-2, 1e-1]$, and the performance on five datasets is presented in Figure 8(b). The results indicate that setting $w_\sigma$ to $1e-3$ or $1e-2$ yields satisfactory outcomes.

## D.3  Time, Memory, and Communication Costs Analysis

As shown in Table 6 and Table 7, we provide an analysis of the time, memory, and communication costs. $A$, $T$, $r$, and $C$ denote the number of agents, the number of tasks, the number of interactions, and the number of classes per task, respectively. $d^o$ and $d$ represent the dimensions of original features and model outputs, respectively. Our method offers five advantages over baselines: shorter runtime, lightweight models, no need for replay, lower memory usage, and reduced interaction cost.

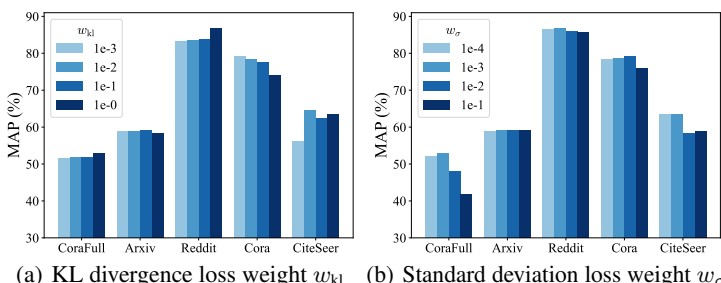

(a) KL divergence loss weight $w_{\mathrm{kl}}$  (b) Standard deviation loss weight $w_\sigma$

Figure 8: The analysis of loss weights.

Table 6: Comparison of time, model parameters, memory size on CoraFull dataset.

| Method | Time cost (s) | Model content | Model params (M) | Memory concent | Memory size (MB) |
|---|---|---|---|---|---|
| Fed-DMSG | **881** | $A$ agents, Server | 33.7 | Replay graph | 79.6 |
| POWER | 1403 | $A$ agents, Server | 20.4 | Replay graph, Global replay graph | 86.2 |
| MASC | 1215 | $A$ agents, Server | 13.5 | Replay graph | 79.6 |
| GHG | 1092 | $A$ agents | **11.5** | Prompts, Classifiers, Prototypes | **7.5** |

Table 7: Comparison of communication costs on CoraFull dataset.

| Method | Complexity | Content | Params (M) |
|---|---|---|---|
| Fed-DMSG | $O(ATrF)$ | Model parameters $F$ | 1885.0 |
| POWER | $O(ATrF + ATCF_{\mathrm{grad}})$ | Model parameters $F$, Class prototype gradients $F_{\mathrm{grad}}$ | 632.5 |
| MASC | $O(ATr(\lvert V\rvert d + h) + AT\lvert V\rvert d^o)$ | Original graph $\lvert V\rvert d^o$, Classifiers $h$, Node embeddings $\lvert V\rvert d$ | 109.4 |
| GHG | $O(AATr((\lvert\widehat{V}\rvert + 2C + 3)d + (\lvert\widehat{V}\rvert + C + 1)d^o + \Phi + h))$ | Prompts $\Phi$, Classifiers $h$, Synthetic graphs $\lvert\widehat{V}\rvert d^o$, Node embeddings $(\lvert\widehat{V}\rvert + 2C + 3)d$, Topology-aware embeddings $(C + 1)d^o$ | **105.0** |

Our approach involves a time cost to facilitate information collaboration among agents, with the goal of enhancing each agent's performance. For each agent at every task of each interaction, the time consumption mainly stems from three aspects: graph-driven organizational structure, customized interactive medium, and prototype-based life cycle. In the graph-driven organizational structure process, the complexity of using SVD to extract the representative subspace is $O((\lvert V\rvert + \lvert\overline{V}\rvert)d^2)$ for the complementarity measure. Similarity calculation is related to the Laplacian smoothing process. As solving for collaborative graph edge weights is a convex optimization problem, it can be quickly solved with convex optimization solvers. Thus, the complexity of graph edge weight calculation is $O((\lvert V\rvert + \lvert\overline{V}\rvert)d^2 + l\lvert\alpha\rvert d^o)$, where $\lvert\alpha\rvert$ returns of the number of edges. Given the number of epochs $E_{\mathrm{syn}}$ on synthetic process, the computational complexity of synthetic graph generation for neighbor agents and task prototype computation are $O(A(l\lvert\alpha\rvert + l\lvert\widehat{V}\rvert + C)d^o E_{\mathrm{syn}})$ and $O(Al\lvert\widehat{V}\rvert d^o + \lvert\overline{V}\rvert d^o)$. It is linear to the number of nodes, the number of edges, and the number of original node attributes.

### D.4 Visualization

As shown in Figure 9, collaborative graph edge weights for the last three tasks on Reddit dataset are displayed. The varying collaborative weights among agents across different tasks reveal autonomous collaborative relationships, which are beneficial for enhanced collaboration and promoting the performance improvement of agents.

## E Discussion

### E.1 Discussion on Related Work

While our approach shares the decentralization and lifelong learning considerations of recent work [38], it differs in several key aspects. Unlike [38], which relies on full-parameter interactions and gradient-based lifelong adaptation, our method explicitly models not only agent similarity but also complementarity and structural relations. Moreover, GHG employs a customized interactive

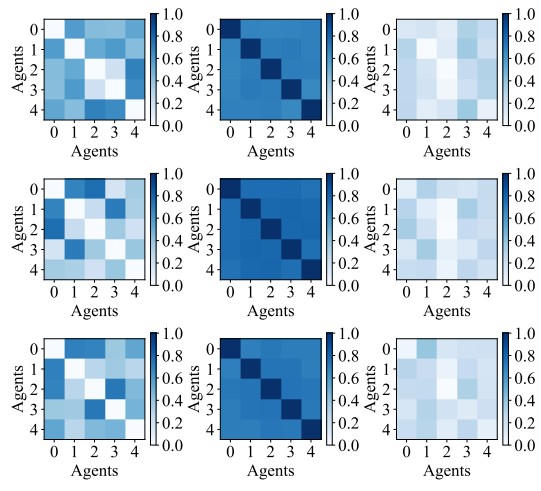

(a) Complementarity Matrix    (b) Similarity Matrix    (c) Collaboration Matrix

Figure 9: The visualization of collaborative graph edge weight matrices on Reddit dataset.

Table 8: Privacy analysis on CoraFull dataset.

| Method | MIA success rate (%) | SRA precision (%) | MAP (%) |
|--------|----------------------|-------------------|---------|
| GHG    | 9.7                  | 1.0               | **54.0** |
| GHG-DP | **4.9**              | **0.9**           | 38.4    |

medium to reduce communication overhead and enhance knowledge compatibility. GHG adopts prototypes and isolation-based strategies to achieve more stable and efficient lifelong learning.

### E.2    Discussion on Privacy Issue

The primary application scenarios of GSL involve cases where multiple laboratories or institutions train models on different graph libraries while aiming to share knowledge collaboratively. Our goal is to share information to enable effective socialized collaboration, unlike graph data across pharmaceutical companies or hospitals, which may not be shareable due to privacy concerns. We still provide a brief discussion on the privacy issue. We conduct Membership Inference Attacks (MIA) and Structure Reconstruction Attacks (SRA) on synthetic graphs transmitted between agents. As shown in Table 8, we report the average MIA success rate and SRA precision across all synthetic graphs. It can be observed that MIA scores are consistently higher than those of SRA. Although synthetic node features are generated based on the original topology, synthetic graphs do not directly transmit topology. Thus, the original structure is well protected. To enhance privacy protection of synthetic graphs, we apply feature perturbation using the Gaussian mechanism [58] to achieve Differential Privacy (DP). As shown in Table 8, both MIA success rate and SRA precision decrease. Although the performance of GHG slightly drops due to DP, it still significantly outperforms baselines.

### E.3    Limitation

For large-scale multi-agent systems operating in complex environments, we employ sparsification techniques to improve scalability. However, when the number of agents grows very large, the current method may incur computational and communication overhead. An interesting direction for future work is to investigate more suitable organizational structures (*i.e.* hierarchical structures) that can better scale to large multi-agent systems. Moreover, we plan to conduct experiments on real-world datasets to further validate the potential of GSL methods in the future.

### E.4    Broader Impact

Our work can positively impact society by contributing to fields that utilize graph-structured data, such as social networks and recommendation systems. It can make information reuse and recommendations more efficient in these fields. However, a lack of regulatory bodies might lead to negative effects.

