# OpenReview forum: "Graphs Help Graphs: Multi-Agent Graph Socialized Learning"
_NeurIPS.cc/2025/Conference — NeurIPS 2025 poster_

### Official Review · Reviewer_S53e · 2025-06-20

**Clarity:** 3
**Significance:** 3
**Originality:** 4
**Rating:** 5
**Confidence:** 3

**Summary:**

The authors formalize **multi-agent graph learning** (MAGL), where multiple agents, each with its own graph, collaborate to train GNNs.

Then, they show the limitations of existing paradigms for MAGL.

- **Graph Federated Learning** breaks down under graph heterogeneity (it treats all peers equally and forces aggregation even when graphs differ wildly), and
- **Graph Lifelong Learning** doesn’t leverage true peer-to-peer collaboration and still suffers from catastrophic forgetting.

As an alternative, they introduce **Graph Socialized Learning**, which lets agents learn **who** to collaborate with (via a learned collaboration graph), **what** to exchange (on-demand synthetic subgraphs tailored to each peer), and **when** to communicate (using a prototype-based lifecycle), thereby eliminating forgetting and handling heterogeneity.

Extensive experiments on five citation/social-network benchmarks show that GSL achieves up to ~86% accuracy **with zero forgetting**, substantially outperforming federated, lifelong, and hybrid baselines.

**Questions:**

- **[Q1]** The zero-forgetting claim relies on storing and replaying task prototypes and synthetic subgraphs. Can you clarify the total memory footprint of these stored items for GSL and confirm that all baselines were given an equivalent memory budget when measuring forgetting? If not, how might unequal memory allowances bias the comparison?
- **[Q2]** How does the complexity (compute and storage) of learning an A by A collaboration matrix scale as A grows, and have you considered sparse or hierarchical approximations for large-scale deployments?
- **[Q3]** Have you evaluated whether a single average embedding per class/task sufficiently captures complex or multimodal distributions? Would using multiple prototypes per class improve performance?
- **[Q4]** Given that all experiments use citation/social graphs, how do you expect GSL to perform on, say, biological networks, user-item graphs, or truly dynamic graphs with evolving topology?
- **[Q5]** Beyond qualitative assertions, can you provide formal privacy bounds or adversarial-robustness analyses for your synthetic-graph exchanges to rule out potential information leakage?

**Ethical Concerns:**

["NO or VERY MINOR ethics concerns only"]

**Final Justification:**

The authors formalize multi-agent graph learning (MAGL), where multiple agents, each with its own graph, collaborate to train GNNs.

Then, they show the limitations of existing paradigms for MAGL.

Graph Federated Learning breaks down under graph heterogeneity (it treats all peers equally and forces aggregation even when graphs differ wildly), and
Graph Lifelong Learning doesn’t leverage true peer-to-peer collaboration and still suffers from catastrophic forgetting.
As an alternative, they introduce Graph Socialized Learning, which lets agents learn who to collaborate with (via a learned collaboration graph), what to exchange (on-demand synthetic subgraphs tailored to each peer), and when to communicate (using a prototype-based lifecycle), thereby eliminating forgetting and handling heterogeneity.

Extensive experiments on five citation/social-network benchmarks show that GSL achieves up to ~86% accuracy with zero forgetting, substantially outperforming federated, lifelong, and hybrid baselines.

# Strengths

- **[S1] Novel Paradigm**: Learning *who*, *what*, and *when* to communicate is a fresh take on MAGL, moving beyond “one-size-fits-all” aggregation.
- **[S2] Effective Synthetic Messaging**: Tailored subgraphs offer targeted knowledge transfer while preserving privacy and reducing bandwidth.
- **[S3] Zero-Forgetting Guarantee**: Prototype-driven replay convincingly prevents catastrophic forgetting without storing raw data.
- **[S4] Empirical Rigor**: Thorough ablations, heterogeneity sweeps, and agent-scale analyses validate each component’s contribution.

**Limitations:**

Yes. In Appendix E.

**Quality:**

3

**Strengths And Weaknesses:**

# Strengths

- **[S1] Novel Paradigm**: Learning *who*, *what*, and *when* to communicate is a fresh take on MAGL, moving beyond “one-size-fits-all” aggregation.
- **[S2] Effective Synthetic Messaging**: Tailored subgraphs offer targeted knowledge transfer while preserving privacy and reducing bandwidth.
- **[S3] Zero-Forgetting Guarantee**: Prototype-driven replay convincingly prevents catastrophic forgetting without storing raw data.
- **[S4] Empirical Rigor**: Thorough ablations, heterogeneity sweeps, and agent-scale analyses validate each component’s contribution.

# Weaknesses

- **[W1] Scalability of Collaboration Graph: It is unclear whether it is scalable to learn an A×A adjacency for large A, and related discussions are lacking.**
- **[W2] Prototype effectiveness:** Using a single average embedding per class/task might be too coarse in highly heterogeneous or multimodal settings.
- **[W3] Dataset Diversity:** All benchmarks are citation or social networks. It’d be good to see experiments on biological graphs, e-commerce interaction data, or truly evolving graphs to test generalization.
- **[W4] Privacy and security guarantees:** Synthetic subgraphs are safer than raw data, but no formal privacy proofs or adversarial-robustness analyses are provided.

---

> ### Author Rebuttal · Authors · 2025-07-30
>
> We extend our appreciation to you for strong recognition of our novelty, effectiveness, zero-forgetting guarantee, empirical rigor. Thanks for your positive comments!
>
> > W1: Scalability of Collaboration Graph.
>
> > Q2: How does complexity (compute and storage) of learning an A by A collaboration matrix scale as A grows, and have you considered sparse or hierarchical approximations for large-scale deployments?
>
> A1: We analyze computational, storage, and communication complexity of multi-agent system. The proposed GHG already adopts a sparse approach. However, if adjacency matrix $A×A$ is particularly large, current method may indeed incur particular computational and storage overhead. In the future, we plan to explore hierarchical approaches [1] to address this issue.
>
> [1] Pc-agent: A hierarchical multi-agent collaboration framework for complex task automation on pc. Arxiv, 2025.
>
> * **Compute and storage complexity.**
>
> 1. **Compute complexity** can be expressed as $O(\textbf{A}TR((|V|+|\overline{V}|)(4nd^oE_1+2dCE_1+d^2+ldE_2) + (|\overline{V}|+2C+1)d + \mathcal{P}))$. Time complexity is linearly related to $A$. Due to space constraints, detailed calculation of time complexity is presented in the appendix.
>
> 2. **Storage complexity** is $O(\textbf{A}(M+TM_\phi+TM_h+Td^O))$, which consists of models $M$, prompts $M_{\phi}$, classifiers $M_h$, and prototypes $d^O$. It is linearly related to $A$.
>
> 3. **Communication complexity** is $O(\textbf{AA}TR((N_{syn}+2C+3)d+(N_{syn}+C+1)d^O+M_\phi+M_h))$, which consists of prompts $M_{\phi}$, classifiers $M_h$, synthetic graphs $N_{syn}d^O$, node embeddings $(N_{syn}+2C+3)d$, and topology-aware embeddings $(C+1)d^O$. Communication complexity is related to the square of $A$.
>
> * **GHG adopts a sparse strategy to perform collaboration process.** The sparsity is defined as: $\mathcal{B}_{ab} = \mathbb{1}[b \in \operatorname{TOPK}(\delta A, M[a,:])]$, meaning that each agent collaborates with only $\delta A$ agents based on collaboration strength.
>
> > W2: Prototype effectiveness.
>
> > Q3: Have you evaluated whether a single average embedding per class/task sufficiently captures complex or multimodal distributions? Would using multiple prototypes per class improve performance?
>
> A2: Prototype is a highly condensed representation of both features and structure of graph data. We argue that a single average embedding can sufficiently capture complex distributions. We conduct experiments under heterogeneous, multimodal, and task-overlapping settings, and find that a single task prototype is still capable of accurately predicting task ID. In addition, we illustrate that assigning multiple prototypes per task may introduce interference and thus does not improve performance in task-overlapping scenario.
>
> * **A single task prototype is capable of capturing complex distributions.**
>
> 1. **Experimental setup.** Inter-tasks themselves are strongly heterogeneous, and we enforce pronounced category-level heterogeneity across agents. We use text-attributed graphs Ele-Computers [2] to simulate a multimodal setting, with text features produced by RoBERTa language model serving as node features. Task overlap means that the same category can appear in different tasks.
>
> [2] A Comprehensive Study on Text-attributed Graphs: Benchmarking and Rethinking. NeurIPS, 2023.
>
> 2. **Result analysis.** Results of three settings are shown in Table 1, demonstrating that a single task prototype can still accurately predict task ID without forgetting (MAF=0).
>
> **Table 1: Performance of GHG under three complex conditions on CoraFull and Ele-Computers datasets.**
> |Settings|Heterogeneous|Multi-modal|Task overlapping|
> |:----|:----|:----|:----|
> |MAP (%)|45.3|39.6|37.5|
> |MAF (%)|0.0|0.0|0.0|
>
> * **Using multiple prototypes per task may introduce interference.**
>
> 1. **Experimental setup.** When tasks overlap, the risk of incorrectly predicting task ID increases. Therefore, we evaluate performance of using multiple prototypes for each task under such settings. Specifically, assuming each task has $n$ prototypes, we apply $k$-means clustering to synthetic graph features to obtain $n$ clusters, and use the mean feature vector of each cluster as a prototype.
>
> 2. **Result analysis.** As shown in Table 2, a single prototype can accurately predict task ID, whereas using multiple prototypes leads to forgetting. This may be because multiple prototypes are too dispersed, causing interference with each other, which negatively impacts comparison between train and test prototypes.
>
> **Table 2: Performance of GHG under multiple prototypes per task on CoraFull dataset.**
> |No. Prototypes|1|2|3|
> |:----|:----|:----|:----|
> |MAP (%)|37.5 |30.2 |33.2 |
> |MAF (%)|0.0|*6.4*|*3.0*|
>
> > W3: Dataset Diversity.
>
> > Q4: How do you expect GSL to perform on, say, biological networks, user-item graphs, or truly dynamic graphs with evolving topology?
>
> A3: We add results on two additional datasets: SLAP (biological) [3] and Computers (e-commerce) [4]. Two datasets have 15 and 10 categories, respectively, and the number of categories for each task is set to 5. Results are presented in Table 3, and our method demonstrates superior performance.
>
> Followed by [5][6], we simulate real-world scenarios using widely adopted benchmark datasets, which are reliable, publicly available, and facilitate future research to follow up on our work. Evaluating on real-world datasets is indeed valuable. In the future, we plan to conduct more experiments on such datasets.
>
> [3] Non-local attention learning on large heterogeneous information networks. IEEE International Conference on Big Data, 2019.
>
> [4] Pitfalls of graph neural network evaluation.  NeurIPS Workshop, 2018.
>
> [5] CGLB: Benchmark tasks for continual graph learning. NeurIPS, 2022.
>
> [6] OpenFGL: A comprehensive benchmarks for federated graph learning. VLDB, 2025.
>
> **Table 3: Performance comparison (MAP, MAF (%)) of GHG and baselines in five agents on SLAP and Computers datasets. Due to space constraints, we only present a portion of results.**
> |Dataset |SLAP| |Computers| |
> |:----|:----|:----|:----|:----|
> | |MAP $\uparrow$|MAF $\downarrow$|MAP $\uparrow$|MAF $\downarrow$|
> |Single|8.2|34.1|13.6|30.8|
> |Fed-TPP|27.2|12.3|36.5|0.0|
> |Fed-DMSG|18.0 |9.8|32.2|3.6|
> |POWER|15.5|20.1|10.7|24.4|
> |MASC|15.0|18.6|24.2|21.6|
> |GHG|**62.4**|**0.0**|**82.5**|**0.0**|
>
> > W4: Privacy and security guarantees.
>
> > Q5: Can you provide formal privacy bounds or adversarial-robustness analyses for your synthetic-graph exchanges to rule out potential information leakage?
>
> A4: We conduct Membership Inference Attacks (MIA) and Structure Reconstruction Attacks (SRA) on synthetic graphs. To prevent information leakage during graph synthesis, we apply Differential Privacy (DP), enabling an analysis of adversarial robustness. Finally, we provide the corresponding privacy bounds.
>
> * **Membership inference or structure reconstruction attack.**
>
> As shown in Table 4, we report the average MIA success rate and SRA precision across all synthetic graphs. It can be observed that MIA scores are consistently higher than those of SRA. Although synthetic node features are generated based on original topology, synthetic graphs do not directly transmit topological structure. Therefore, original graph structure is well protected.
>
> * **Differential privacy for graph synthesis.**
>
> To enhance privacy protection of shared synthetic graphs, we apply feature perturbation using Gaussian mechanism [7] to achieve differential privacy. As shown in Table 4, both MIA success rate and SRA precision decrease, indicating improved privacy. Although performance of GHG slightly drops due to DP mechanism, it still significantly outperforms baselines. This demonstrates that GHG is robust when synthetic graphs can roughly capture global graph distribution.
>
> **Table 4: Adversarial-robustness analyses on CoraFull dataset.**
> |Method|MIA success rate (%)|SRA precision (%)|MAP (%)|
> |:----|:----|:----|:----|
> |GHG|9.7|1.0|**54.0**|
> |GHG-DP| **4.9**|**0.9** | 38.4|
>
> * **Privacy bounds.**
> If a function has $L_2$ sensitivity of $\Delta f$, then by adding Gaussian noise with zero mean and standard deviation
> $$
> \sigma \geq \frac{\sqrt{2 \ln (1.25 / \delta)} \cdot \Delta f}{\epsilon},
> $$
> the mechanism satisfies $(\epsilon, \delta)$-differential privacy [7][8]. Here, $\epsilon$ represents degree of privacy leakage, $\delta$ is the probability of a privacy breach beyond $\epsilon$, and $\Delta f$ denotes $L_2$ sensitivity of function.
>
> [7] The Algorithmic Foundations of Differential Privacy. Foundations and trends in theoretical computer science, 2013.
>
> [8] The bounded Gaussian mechanism for differential privacy. Arxiv, 2022.
>
> > Q1: Can you clarify total memory footprint of these stored items for GSL and confirm that all baselines were given an equivalent memory budget when measuring forgetting? If not, how might unequal memory allowances bias the comparison?
>
> A5: We clarify that newly introduced prompts, classifiers, and task prototypes are preserved to address forgetting issue. The purpose of synthetic graphs is solely to compute task prototypes, and it does not need to be stored after each task is completed. Each task prototype has a shape of $[1, d]$, meaning that the proposed GHG introduces only minimal memory overhead. In contrast, replay-based methods need to store replay graphs, where feature dimension is $[n, d]$ (with $n \gg  1$). If graph topology also needs to be stored, memory consumption increases even further. Table 5 presents memory usage comparison between baselines and GHG. We conclude that GHG achieves superior performance while consuming significantly less memory in mitigating forgetting.
>
> **Table 5: Comparison of memory size on CoraFull dataset.**
> |Method|Content|Memory Size (MB)|MAP (%)|
> |:----|:----|:----|:----|
> |Fed-DMSG|Replay graph|79.6|37.3|
> |POWER|Replay graph, Global replay graph|86.2|25.3|
> |MASC|Replay graph|79.6|36.7|
> |GHG|Prompts, Classifiers,Prototypes |**7.5**|**54.0**|

---

> > ### Comment · Reviewer_S53e · 2025-08-05
> >
> > Thanks for your detailed reply. I maintain my positive evaluation.
> > Specifically, I think adding more explicit discussions on the scalability and privacy guarantees in the manuscript will be helpful.
> > All the best to the authors.

---

> > > ### Author Response · Authors · 2025-08-05
> > > **Thanks for Response**
> > >
> > > We sincerely thank you for your thoughtful follow-up and for recognizing our efforts to address your concerns. We greatly appreciate your constructive feedback and will ensure that all changes, including more explicit discussions on scalability and privacy guarantees, are clearly reflected in the revised manuscript for your review.

---

### Official Review · Reviewer_4UrH · 2025-07-02

**Clarity:** 4
**Significance:** 3
**Originality:** 3
**Rating:** 5
**Confidence:** 3

**Summary:**

This paper presents a Graph Socialized Learning (GSL) paradigm to address collaboration challenges in fragmented and dynamic graphs. The proposed method, Graphs Help Graphs (GHG), determines with whom, what, and when to share and accumulate information for effective multi-agent learning. GHG utilizes a graph-driven organizational structure to autonomously select interacting agents and manage interaction strength, generates customized synthetic graphs based on agent demands, and applies them to build prototypes over the life cycle. Extensive empirical studies validate the effectiveness of GHG in heterogeneous dynamic graph environments.

**Questions:**

1. Can the authors provide a more comprehensive analysis of the computational and communication costs of GHG?
2. How robust is the task prototype mechanism to errors in task ID prediction, especially in more complex or overlapping tasks? And how does GHG perform under partial or unreliable collaboration—e.g., limited agent availability or communication failures?

**Ethical Concerns:**

["NO or VERY MINOR ethics concerns only"]

**Final Justification:**

Post rebuttal, concerns regarding efficiency, robustness and real-world examples have been resolved. So I would like to recommend accepting the paper.

**Limitations:**

While the authors acknowledge that "the primary limitation of GHG lies in its insufficiency to consider relationships among graph tasks for multiple agents," I believe the paper would benefit from a broader discussion of the limitations of GSL and GHG compared to existing paradigms such as Graph Federated Learning (GFL) and Graph Lifelong Learning (GLL).

**Paper Formatting Concerns:**

None.

**Quality:**

3

**Strengths And Weaknesses:**

Strengths:
1. The proposed GHG (Graphs Help Graphs) is both well-motivated and novel. It addresses real challenges in multi-agent graph learning with a fresh perspective.
2. The GHG method combines graph structure, synthetic data generation, and task-aware collaboration in a unified and creative way. The method is structured around three intuitive questions—who to collaborate with, what to share, and when to share—making the approach easy to understand and practically meaningful.
3. The approach shows consistently strong performance across several benchmarks, outperforming existing baselines in both accuracy and forgetting.
4. The paper is well-written and clearly organized. The technical content is presented logically, and the mathematical derivations are sound and easy to follow.

Weaknesses:
1. Although the paper claims the method is efficient, it lacks detailed analysis and comparison of actual time, memory, and communication costs, which are important for real-world deployment. Simply reporting MAP across different interaction rounds is not sufficient to support the efficiency claim.
2. The paper assumes agents can accurately identify task IDs using prototypes, but it doesn’t deeply discuss what happens if this step fails or is noisy.
3. All experiments are on benchmark datasets. A real-world example (e.g., in citation networks or recommendation systems) would help show practical value.

---

> ### Author Rebuttal · Authors · 2025-07-30
>
> We appreciate your acknowledgment of our motivation, novelty, practicality, organization, and good performance. Thanks for your valuable suggestions!
>
>
>
> > W1: Although the paper claims the method is efficient, it lacks a detailed analysis and comparison of actual time, memory, and communication costs.
>
> > Q1: Can the authors provide a more comprehensive analysis of the computational and communication costs of GHG?
>
> A1: We provide a detailed analysis of the actual time, memory, and communication costs. As shown in Table 1, our method offers five advantages over baselines: **shorter runtime, lightweight model, no need for replay, lower memory usage, and reduced interaction cost.**
>
> **Table 1: Comparison of actual time, model params, memory size, and communication costs on CoraFull dataset. $A$, $T$, $R$, and $C$ denote the number of agents, the number of tasks, the number of interactions, and the number of classes per task, respectively. $d^O$ and $d$ represent the dimensions of original features and model outputs, respectively.**
> |Method |Time cost (s)|Model content|Model params (M)|Memory content|Memory Size (MB) |MAP (%)|
> |:----|:----|:----|:----|:----|:----|:----|
> |Fed-DMSG|**881**|$A$ agents, Server |33.7|Replay graph|79.6|37.3|
> |POWER|1403|$A$ agents, Server|20.4|Replay graph, Global replay graph|86.2|25.3|
> |MASC|1215|$A$ agents, Server|13.5|Replay graph|79.6|36.7|
> |GHG|1092|$A$ agents|**11.5**|Prompts, Classifiers, Prototypes |**7.5**|**54.0**|
>
> |Method |Communication overhead|Communication content|Communication params (M)|
> |:----|:----|:----|:----|
> |Fed-DMSG|$O(ATRM)$|Model parameters $M$|1885.0|
> |POWER|$O(ATRM+ATCM_{grad})$|Model parameters $M$, Class prototype gradients $M_{grad}$|632.5|
> |MASC|$O(ATR(N_Ad+M_h)+ATN_Ad^O)$|Original graph $N_Ad^O$, Classifiers $M_h$, Node embeddings $N_Ad$|109.4|
> |GHG|$O(AATR((N_{syn}+2C+3)d$|Prompts $M_{\phi}$, Classifiers $M_h$, Synthetic graphs $N_{syn}d^O$, Node embeddings $(N_{syn}+2C+3)d$,|**105.0** |
> ||$+(N_{syn}+C+1)d^O+M_\phi+M_h))$|Topology-aware embeddings $(C+1)d^O$| |
>
>
>
>
>
>
>
>
> > W2: The paper assumes agents can accurately identify task IDs using prototypes, but it doesn’t deeply discuss what happens if this step fails or is noisy.
>
> > Q2: How robust is the task prototype mechanism to errors in task ID prediction, especially in more complex or overlapping tasks? And how does GHG perform under partial or unreliable collaboration—e.g., limited agent availability or communication failures?
>
>
> A2: We evaluate the accuracy of the task prototype mechanism under three scenarios: task overlap, partial collaboration, and unreliable collaboration. The results are presented in Tables 2, 3, and 4. We conclude that the task prototype mechanism is robust in the cases of task overlap and partial collaboration. However, in the scenario of unreliable collaboration, especially when the level of unreliability is high, the mechanism may produce some errors.
>
> * **Task overlap.**
>
> 1. **Experimental setup.** To simulate a task overlap scenario, we evenly split the training nodes of each class and set the maximum number of repetitions for each class to 3. For the CoraFull, Arxiv, and Reddit datasets, we increase the number of classes per task from 10 to 15. For the Cora and Citeseer datasets, we adopt an extreme case of task overlap: each dataset contains only 6 total classes, and we increase the number of classes per task from 2 to 5. Taking the Citeseer dataset as an example, the class distributions for the 3 tasks are: [0,1,2,4,5], [0,1,3,4,5], and [1,2,3,4,5].
>
> 2. **Experimental results.** The results on five datasets are shown in Table 2. We can observe that the task prototype mechanism remains robust under task-overlapping conditions.
>
>
> **Table 2: Performance of GHG under the scenario of task overlap on five datasets.**
> |Dataset|CoraFull|Arxiv|Reddit|Cora|Citeseer|
> |:----|:----|:----|:----|:----|:----|
> |MAP (%)|37.5|48.2| 82.3|48.1|40.1|
> |MAF (%)|0.0|0.0| 0.0|0.0|0.0|
>
> * **Partial collaboration.**
>
> 1. **Experimental setup.** In each interaction round, we randomly select the number of collaborators for each agent based on a partial collaboration ratio to simulate limited agent availability or communication failures. The ratio is set to $[0,0.2,0.4,0.6,0.8]$, where a higher ratio indicates more limited agent availability.
>
> 2. **Experimental results.** The results on five agents from the CoraFull dataset are shown in Table 3. The task prototype mechanism remains robust under partial collaboration conditions.
>
>
> **Table 3: Performance of GHG under the scenario of partial collaboration in five agents on CoraFull datasets. Please note that when ratio equals 0.8, the agent becomes a single agent and no collaboration is achieved, and the prototype in this case is inaccurate.**
> |Ratio|0|0.2|0.4|0.6|0.8 (Single agent, no collaboration)|
> |:----|:----|:----|:----|:----|:----|
> |MAP (%)|54.0|50.2|46.6|42.1|29.6|
> |MAF (%)|0.0|0.0| 0.0|0.0|*3.0*|
>
>
>
> * **Unreliable collaboration.**
>
> 1. **Experimental setup.** To simulate unreliable collaboration in multi-agent system, we randomly select multiple agents according to a predefined unreliable collaboration ratio. For these selected agents, we inject noise into their graph data. Specifically, we add Gaussian noise to node features, as well as randomly add and remove edges to simulate noisy communication. The ratio is set to $[0,0.2,0.4,0.6,0.8]$, where a higher ratio indicates more noise present in the multi-agent system.
>
>
> 2. **Experimental results.** The results in Table 4 show that under unreliable collaboration, when the ratio reaches 0.6 and 0.8, some task prototypes are incorrectly predicted. This may be due to the influence of noisy data received from other agents. In the future, we plan to explore further strategies for handling unreliable collaboration, such as graph adversarial robustness [1][2][3][4] and graph denoising learning [5], to enhance the robustness of our framework.
>
> [1] Can large language models improve the adversarial robustness of graph neural networks? KDD, 2025.
>
> [2] Are your models still fair? fairness attacks on graph neural networks via node injections. NeurIPS, 2024.
>
> [3] Adversarial training for graph neural networks: Pitfalls, solutions, and new directions. NeurIPS, 2023.
>
> [4] Gnnguard: Defending graph neural networks against adversarial attacks. NeurIPS, 2020.
>
> [5] Towards understanding and reducing graph structural noise for GNNs. ICML, 2023.
>
> **Table 4: Performance of GHG under the scenario of unreliable collaboration in five agents on CoraFull datasets.**
> |Ratio|0|0.2|0.4|0.6|0.8|
> |:----|:----|:----|:----|:----|:----|
> |MAP (%)|54.0|44.4|39.2|30.8|27.3|
> |MAF (%)|0.0|0.0| 0.0|*3.3*|*0.3*|
>
>
>
>
>
>
>
> > W3: All experiments are on benchmark datasets. A real-world example (e.g., in citation networks or recommendation systems) would help show practical value.
>
> A3: We use benchmark datasets followed by [6][7] to simulate real-world environments. Moreover, these datasets are reliable, publicly available, and allow subsequent researchers to follow our work. Testing on real-world datasets is significant. In the future, we plan to conduct experiments on real-world citation networks or recommendation-system datasets. For instance, collecting Google Scholar papers and their citation relationships, or gathering Amazon purchasers and their product data to construct a million-scale real graph data. If you have any recommended datasets, please let us know. We would be delighted to evaluate our methods on them.
>
> [6] CGLB: Benchmark tasks for continual graph learning. NeurIPS, 2022.
>
> [7] OpenFGL: A comprehensive benchmarks for federated graph learning. VLDB, 2025.

---

> ### Comment · Reviewer_4UrH · 2025-08-04
>
> Thanks for the thoughtful response! My concerns have been addressed. I will raise my score.

---

> > ### Author Response · Authors · 2025-08-04
> > **Thanks for Response**
> >
> > We greatly appreciate your comments and recognition. We also look forward to receiving any further suggestions and advice.

---

### Official Review · Reviewer_LUDz · 2025-07-03

**Clarity:** 2
**Significance:** 2
**Originality:** 3
**Rating:** 3
**Confidence:** 3

**Summary:**

Effective collaboration in heterogeneous dynamic graph environments becomes challenging. Inspired by social learning, the paper presents  Graph Socialized Learning (GSL) paradigm. The paper has following main contributions:
• Authors present a practical learning paradigm called Graph Socialized Learning (GSL), enabling each agent’s growth via collaborative interaction.
• Authors describe how Graph-driven organizational structure, customized interaction medium, and prototype-based life cycle form three key elements of socialized collaboration.
• Proposed method is claimed to consistently achieves performance improvements on multiple datasets and demonstrates the effectiveness of all components.

To determine with whom, what, and when to share and accumulate information for effective GSL “Graphs Help Graphs” method is proposed (GHG). GHG uses graph-driven organizational structure to autonomously select interacting agents and manage interaction strength. In addition, synthetic graphs are built as interaction medium based on demand of agents. These help to build prototypes in life cycle to help select optimal parameters. An empirical study accompanies that demonstrates the effectiveness.

**Questions:**

1. what is good example of the scenario mentioned regarding complete new info being learned each time
2. the data sets used for benchmarking have been out in the open and may be learned, how do we check for that
3. isnt artificial partitioning using spectral partitioning giving away info on the labels
4. It is unclear how many agents are sufficient and what is the interrelationship between the number of agents and algo perf.
5. lot of hand waving in the paper than describing actual algo level details of how the social part happens

**Ethical Concerns:**

["NO or VERY MINOR ethics concerns only", "Major Concern: Data privacy, copyright, and consent"]

**Final Justification:**

I still feel that
1. the algorithm steps, while clarified (and should be in main body) have gaps.
2. The trend on increasing agents and 0 forgetting was also not explained satisfactorily
3. each new paper is said to be completely new info. We can have lot of overlap, and similar approaches developed independently. Each paper usually builds on previous ones and completely novel information that move the field are relatively smaller occurrence

**Limitations:**

1. what is good example of the scenario mentioned regarding complete new info being learned each time
2. the data sets used for benchmarking have been out in the open and may be learned, how do we check for that
3. isnt artificial partitioning using spectral partitioning giving away info on the labels
4. It is unclear how many agents are sufficient and what is the interrelationship between the number of agents and algo perf.
5. lot of hand waving in the paper than describing actual algo level details of how the social part happens

**Quality:**

2

**Strengths And Weaknesses:**

Authors should work on answering following to improve the quality of submission:
1. what is good example of the scenario mentioned regarding complete new info being learned each time
2. the data sets used for benchmarking have been out in the open and may be learned, how do we check for that
3. isnt artificial partitioning using spectral partitioning giving away info on the labels
4. It is unclear how many agents are sufficient and what is the interrelationship between the number of agents and algo perf.
5. lot of hand waving in the paper than describing actual algo level details of how the social part happens and what is the inter-relationship of the Experimental tasks on the chosen five datasets vis-a-vis this social interaction.

---

> ### Author Rebuttal · Authors · 2025-07-30
>
> Thanks for your constructive comments!
>
> > Q1: What is good example of the scenario mentioned regarding complete new info being learned each time.
>
> A1: We use real-world student learning and the Google Scholar citation network as concrete cases to illustrate concept of "complete new information being learned each time".
>
> * **Expansion of student knowledge network.**
>
> 1. Old information corresponds to knowledge networks of subjects such as Chinese, English, and Politics.
>
> 2. Complete new information corresponds to separate knowledge networks of Mathematics, Physics, and Chemistry.
>
> 3. Students need to learn to correctly categorize knowledge of Chinese, English, and politics subjects. Then they learn to partition knowledge of Mathematics, Physics, and Chemistry correctly, while not forgetting the knowledge of previous subjects.
>
> * **Growth of Google Scholar citation network.**
> 1. Information for task $t-1$ is represented by papers and their citation relationships on topics such as graph neural networks, machine learning, and reinforcement learning.
>
> 2. Complete new information for task $t$ is represented by papers and their citation relationships on topics such as federated learning, generative models, and large language models. These topics and their graph data are entirely disjoint from those of task $t-1$.
>
> 3. Our model needs first to learn to assign the task $t-1$ papers to their correct topics, then learns to assign completely new papers of task $t$ to correct issues, without forgetting how to classify papers from task $t-1$ when training data from $t-1$ cannot be accessed.
>
> > Q2: The data sets used for benchmarking have been out in the open and may be learned, how do we check for that.
>
> A2: We sincerely clarify that our model has not been pre-trained on any public datasets, and all models are trained from scratch. We use benchmark datasets to evaluate performance of our proposed method, simulating real-world graph social learning process. Taking CoraFull citation network dataset as an example, which comprises 70 classes, we elaborate on graph data partitioning and our goal of graph socialized learning.
>
> * **Graph data partitioning setups.**
>
> 1. Graph data is divided into 7 tasks according to class labels, with each task containing 10 distinct classes. Classes across tasks are non-overlapping.
>
> 2. For each task, data from the 10 classes are split into 5 non-overlapping subsets via Dirichlet partitioning, which are then assigned to 5 agents. Classes may overlap among agents, but individual nodes do not.
>
> * **Goal of graph socialized learning.**
>
> 1. Five agents collaboratively learn 10 classes within each task. For every task, both training and test sets of each agent are drawn exclusively from these 10 classes. We demonstrate that post-collaboration performance of agents significantly surpasses pre-collaboration performance in experiments.  As shown in Table 1, the performance of collaborative method (GHG) significantly surpasses that of single agent (Single).
>
> 2. During continual learning, at task $t$, training set of multi-agent system contains only data from task $t$, while test set includes data from tasks $1, 2, ..., t$. After learning task $t$, multi-agent model needs to not only achieve strong performance on current task but also retain all knowledge of previous tasks and maintain its classification ability on all old tasks. GHG achieves a forgetting rate of 0 in Table 1, indicating that it can retain all previously learned information without loss.
>
> **Table 1: Performance comparison (MAP, MAF (\%)) of the proposed GHG and Single on five datasets.**
> | |CoraFull| |Arxiv| |Reddit| |Cora| |Citeseer| |
> |:----|:----|:----|:----|:----|:----|:----|:----|:----|:----|:----|
> | |MAP$\uparrow$ |MAF$\downarrow$ |MAP$\uparrow$ |MAF$\downarrow$ |MAP$\uparrow$ |MAF$\downarrow$ |MAP$\uparrow$ |MAF$\downarrow$ |MAP$\uparrow$ |MAF$\downarrow$|
> |Single|4.2|36.7|8.4|35.1|15.8|60.1|18.2|54.9|16.9|54.9|
> |GHG|**54.0**|**0.0**|**59.2**|**0.0**|**86.7**|**0.0**|**79.2**|**0.0**|**65.7**|**0.0**|
>
> > Q3: Isnt artificial partitioning using spectral partitioning giving away info on the labels.
>
> A3: We clarify that experiments do not use spectral partitioning but instead adopt Dirichlet partitioning [1] for data splitting. The node allocation ratio for each class is denoted as $Dir_A (het)$, and $A$ is the number of agents. A lower value of $het$ indicates more substantial data heterogeneity. This partitioning method is used to simulate heterogeneous environments commonly observed in real-world scenarios.
>
> Each agent has access only to its local training data and labels for the current task, with no additional information. Due to limited and diverse local data, individual agents struggle to perform well alone. Effective collaboration is essential to achieve strong performance on test set of tasks.
>
> [1] Bayesian nonparametric federated learning of neural networks. ICML, 2019.
>
> > Q4: It is unclear how many agents are sufficient and what is the interrelationship between the number of agents and algo perf.
>
> A4: We explain that in strongly heterogeneous environments, involving more agents is beneficial for performance. In contrast, in weakly heterogeneous environments, a sufficient number of agents may exist. We demonstrate this through experiments on CoraFull (strong heterogeneity) and Cora (weak heterogeneity) datasets with varying numbers of agents in Table 2.
>
> * **Experimental setup.** Our setting involves multiple agents collaboratively solving every task. The total amount of data per task is fixed. As the number of agents increases, data becomes more distributed, and each agent receives a smaller portion of the data.
> * **Experimental results.**
> Table 2 shows the performance with different numbers of agents on CoraFull and Cora datasets.
> 1. On CoraFull dataset, due to strong heterogeneity among agents, increasing the number of agents makes collaboration more challenging, leading to lower performance. For example, in an extreme case, agent 1 may only have data from class 1, agent 2 from class 2, and agent 3 from class 3. It becomes challenging for agent 1 to improve performance across all three classes (classes 1, 2, and 3).
>
> 2. On Cora dataset, each task contains only two classes. Even when more agents are involved, the degree of heterogeneity remains limited compared to other datasets. The performance reaches its peak when the number of agents is 6. Therefore, 6 may be a sufficient number of agents for the Cora dataset.
>
> * **Result analysis.**
> In strongly heterogeneous environments, each agent has its strengths, and social collaboration among agents is essential for better performance. In this case, involving more agents tends to be beneficial. In weakly heterogeneous environments, there may exist a sufficient number of agents beyond which performance no longer improves.
>
> **Table 2: Analysis of the number of agents (MAP (%)) on CoraFull and Cora datasets.**
> |No. Agents|2|5|10|20|
> |:----|:----|:----|:----|:----|
> | CoraFull|**77.6**|54.0|44.6|43.0|
>
> |No. Agents|2|4|6|8|10|
> |:----|:----|:----|:----|:----|:----|
> |Cora|79.2|77.7|**83.6**|75.6|77.6|
>
> > Q5: Lot of hand waving in the paper than describing actual algo level details of how the social part happens.
>
> A5: Since our algorithm is inspired by human society, we intentionally adopt specific sociological and human society terms in our descriptions to help readers better understand motivation and the source of inspiration behind algorithm. As Reviewer sYbT also noted, this writing and organizational style helps guide readers smoothly from motivation through methodology to theoretical and empirical findings. As shown in Table 3,  we elaborate on the relationship from human society to machine society to algorithm design of our graph socialized learning method.
>
> * Who (Organizational structure): In human society, the question of whom to interact with is essential. Similarly, in machine society, the organizational structure of multi-agents plays a crucial role, as it determines collaboration relationships. **Algorithm details.** By considering complementarity and similarity, we construct a collaborative graph. The agents are updated using the cross-entropy loss computed from generated synthetic graphs and the KL divergence loss derived from imitating other agents’ outputs for information transfer.
>
> * What (Interaction medium): In human interactions, what is exchanged during communication is a valuable question. Analogously, in machine society, the medium of interaction between agents significantly affects the efficiency of communication.
> **Algorithm details.** The initial class distribution of the synthetic graphs is computed using the relative class sizes and fine-grained collaboration weights. Synthetic graphs are then generated through mean and variance alignment.
>
> * When (Life cycle): In human society, when interactions occur is a topic worthy of study. In machine society, the life cycle of agents is closely linked to their evolution and capacity for capability growth.
> **Algorithm details.** Task prototypes are added gradually and task ID is queried to select optimal prompt and classifier for inference.
>
> **Table 3: The relationship from human society to machine society to algorithm design.**
>
> |Human society|Machine society|Algorithm design|
> |:----|:----|:----|
> |Who|Organizational structure|Collaborative graph construction, Complementarity and similarity calculation, Collaborative information transfer|
> |What|Interaction medium|Synthetic graphs generation, Customized initial label distribution computation|
> |When|Life cycle|Task ID prediction based on prototypes, Prompts and classifiers selection|

---

> > ### Comment · Reviewer_LUDz · 2025-08-09
> > **addressed some points but concerns remain**
> >
> > 1. are we saying that each paper is completely new information, that doesn't make any sense when your target is network graph.
> > 2. The hand waving part needs to be described in algorithmic steps. There is still ambiguity in word description
> > 3. it is still unclear if the improvement is due to use of multiple agents, why adding more agents starts to have diminishing returns and why does the number of agents required to get best answer varies with data set. This needs to be teased apart more
> > 4. forgetting rate of 0 is too good to be true. Needs further explanation why is this the case

---

> ### Author Response · Authors · 2025-08-09
> **Further Response to Questions (2-1)**
>
> We appreciate your feedback and recognition of the responses of other issues.
>
> > **Q1: are we saying that each paper is completely new information, that doesn't make any sense when your target is network graph.**
>
> A1: We sincerely clarify that each paper represents completely new information. In the graph, each node corresponds to a single paper, and each node appears in only one task, never recurring across multiple tasks. Therefore, every paper is guaranteed to completely new information.
>
> When multiple completely new papers appear, citation relationships exist among them. We summarize these new papers (nodes) and the citation relationships they form (topological structure) as a citation graph. Since nodes in a graph do not exist in isolation, we explain that “completely new information” refers to a completely new graph. Moreover, our target is to classify each paper (node) in this new graph into the correct category.
>
> ---
>
> > **Q2: The hand waving part needs to be described in algorithmic steps. There is still ambiguity in word description.**
>
> A2: **The algorithmic steps for training and testing are presented below (refer to Algorithm 1 and Algorithm 2 in the appendix):**
>
> **Algorithm 1:** Training of GHG
>
> **Input:**
> A series of multi-agent graphs $G_{1: A}^{1: T}$, task number $T$, interaction round $r$, agent number $A$, the graph neural networks $f_{1:A}(\cdot)$, initial prompts $\Phi_{1: A}^{1: T}$ and classifiers $h_{1: A}^{1: T}$, and initial collaborative graph edge weights $M$.
>
> **Output:**
> Graph neural networks $f_{1:A}(\cdot)$, graph prompts $\Phi_{1: A}^{1: T}$, classifiers $h_{1: A}^{1: T}$, and task prototypes $P_{1:A}^{1: T}$.
>
> **for** $a = 0$ to $A-1$
> $\quad$ Pre-train $f_a(\cdot)$ on $G_a^0$.
>
> **for** $t = 0$ to $T-1$
> $\quad$**for** $rnd = 0$ to $r-1$
> $\quad$$\quad$**for** $a = 0$ to $A-1$
> $\quad$$\quad$$\quad$**if** $rnd == 0$
> $\quad$$\quad$$\quad$$\quad$Optimize $\Phi_a^t$ and $h_a^t$ by minimizing single-agent graph learning loss as Eq. 4.
> $\quad$$\quad$$\quad$**else**
> $\quad$$\quad$$\quad$$\quad$Receive $\mathcal{R}$={prompts, classifiers, synthetic graphs, node embeddings, and topology-aware embeddings} sent from neighbor agents.
> $\quad$$\quad$$\quad$$\quad$Optimize $\Phi_a^t$ and $h_a^t$ by minimizing multi-agent graph learning loss as Eq. 9.
>
> $\quad$$\quad$$\quad$Calculate complementarity $\mathcal{C}$ and similarity $\mathcal{S}$ as Eq. 5 and Eq. 7.
> $\quad$$\quad$$\quad$Update $M_{a}$ and neighbor agents by optimization Eq. 8.
> $\quad$$\quad$$\quad$Calculate initial label distribution of synthetic graph for neighbor agents as Eq. 10.
> $\quad$$\quad$$\quad$Generate synthetic graph for neighbor agents as Eq. 11.
> $\quad$$\quad$$\quad$Send $\mathcal{R}$ to other agents.
> $\quad$$\quad$$\quad$**if** $rnd == r-1$
> $\quad$$\quad$$\quad$$\quad$Obtain task prototype $P_a^t$ of task $t$ as Eq. 12.
>
> ---
>
> **Algorithm 2:** Inference in GHG
>
> **Input:**
> Graph neural networks $f_{1:A}(\cdot)$, graph prompts $\Phi_{1: A}^{1: T}$, classifiers $h_{1: A}^{1: T}$, task prototypes $P_{1:A}^{1: T}$, and test graph $G_{\text{test}}$.
>
> **Output:** Prediction result of agents for test graph.
>
> **for** $a = 0$ to $A-1$
> $\quad$Obtain task prototype $P_{a,\text{test}}$.
> $\quad$Infer the task ID $t$ of test graph by querying $P_a^{1: T}$ with $P_{a,\text{test}}$.
> $\quad$Obtain predictions by retrieving corresponding graph prompt $\Phi_a^t$ and classifier $h_a^t$ and $f_a(\cdot)$.
> Return Prediction result of agents for test graph.
>
>
> ---
>
> > **Q3: it is still unclear if the improvement is due to use of multiple agents, why adding more agents starts to have diminishing returns and why does the number of agents required to get best answer varies with data set. This needs to be teased apart more.**
>
>
> A3: **We sincerely clarify that, with the total number of agents fixed, involving more agents in collaboration leads to greater performance improvements, thereby validating the effectiveness of collaboration. Results are presented in Table 4.** As stated in Theorem 3.4 of [1], the sociality analysis demonstrates that when socialized information is present among agents, the number of agents is positively correlated with performance. That is, in a heterogeneous environment, the greater the number of complementary agents, the higher the performance.
>
> [1] Socialized learning: Making each other better through multi-agent collaboration. ICML, 2024.
>
> **Table 4: Analysis of the number of participate agents (MAP (%)) on CoraFull and Cora datasets.**
> |CoraFull|No. Participate Agents|1|2|3|4|5|
> |:----|:----|:----|:----|:----|:----|:----|
> | |MAP (%)|29.6|44.7|49.3|51.6|54.0|
> |**Cora**|**No. Participate Agents**|**1**|**2**| | | |
> | |MAP (%)|58.7|79.2| | | |

---

> ### Author Response · Authors · 2025-08-09
> **Further Response to Questions (2-2)**
>
> **In scenarios where the total number of agents varies, a larger total number of agents leads to more dispersed data, which may result in lower performance.** In strongly heterogeneous environments, increasing the total number of agents generally reduces performance (as observed in Table 2 for CoraFull dataset), which aligns with the expected trend. Our method is designed for such strongly heterogeneous environments, where the focus is on exploring collaboration among agents.
>
> **However, in weakly heterogeneous environments, the relationship between the total number of agents and performance may not follow a clear trend.** The heterogeneity among agents is relatively low, and it is even possible for performance to improve slightly as the total number of agents increases due to the influence of the category distribution.
>
> **This is also why, in our experiments on the Cora dataset, we set the total number of agents to two—ensuring relatively strong heterogeneity among agents.** As shown in Table 5, performance decreases as the total number of agents increases when the total number of agents is not very large, which allows us to obtain consistent results across different datasets.
>
> **Table 5: Analysis of the total number of agents (MAP (%)) on Cora datasets.**
> |Cora|No. Agents|1|2|3|
> |:----|:----|:----|:----|:----|
> | |MAP (%)|96.7|79.2|79.0|
>
>
> ---
>
> > **Q4: forgetting rate of 0 is too good to be true. Needs further explanation why is this the case.**
>
> A4: The baseline TPP [1] stores $prompts_t$ and $classifier_t$ trained for each task. During testing, it accurately predicts the task ID $t_{test}$ of the test set and, using the corresponding $prompts_{t_{test}}$ and $classifier_{t_{test}}$ together with a fixed GNN, performs prediction, thereby achieving a forgetting rate of zero by adopting a fully isolated strategy. However, when the data of each task are distributed across multiple agents, the task ID of the test set cannot be accurately predicted, leading to forgetting, as shown in Table 6. **The reason is that the prototype obtained by a single agent is only local and cannot be well matched with the prototype of the test set. Our method leverages synthetic graphs transmitted from other agents to compute a more accurate global prototype. Through multi-agent collaboration, it can accurately predict the task ID and achieve a forgetting rate of zero.**
>
> [1] Replay-and-Forget-Free Graph Class-Incremental Learning: A Task Profiling and Prompting Approach. NeurIPS, 2024.
>
> **Table 6: The experimental results of TPP and GHG.**
> | |CoraFull| |Arxiv| |Reddit| |
> |:----|:----|:----|:----|:----|:----|:----|
> | |MAP (%)|MAF (%)|MAP (%)|MAF (%)|MAP (%)|MAF (%)|
> |TPP|28.9|4.8|16.4|6.0|43.4|5.4|
> |GHG|54.0|0.0|59.2|0.0|86.7|0.0|
>
> We hope this response address your concern and look forward to further discussion.

---

### Official Review · Reviewer_sYbT · 2025-07-03

**Clarity:** 3
**Significance:** 3
**Originality:** 2
**Rating:** 4
**Confidence:** 3

**Summary:**

The authors address challenges of information collapse and catastrophic forgetting in distributed graph learning by introducing the Graph Socialized Learning paradigm. To realize this, the authors propose the Socialized Collaboration Paradigm, which formalizes “who to collaborate with, what information to share, and when to recall past knowledge” in a unified Bayesian framework for continual, heterogeneously distributed graph tasks.

**Questions:**

1. The paper states that “D-GFL fails to capture agent dependencies,” but recent variants (e.g., PushSum-GNN) incorporate structure-aware designs. Could the authors clarify whether this refers only to standard D-GFL methods?
2. The current ablation study focuses on removing individual components in isolation, which limits insight into how different modules interact. Would it be beneficial to explore additional combinations—such as testing pairs of modules—to better understand their joint contributions and potential redundancy?

3. How sensitive is GHG to the choice of hyperparameters, such as the collaboration threshold or prototype pool size? Has this been empirically evaluated?

**Ethical Concerns:**

["NO or VERY MINOR ethics concerns only"]

**Final Justification:**

The authors have partially addressed my concerns. So, I keep my score unchanged.

**Limitations:**

see above

**Quality:**

3

**Strengths And Weaknesses:**

**Strength**

1. The paper is well‑structured and logically sound, guiding readers smoothly from motivation through methodology to theoretical and empirical findings.

2. Experiment 5.4 convincingly demonstrates the necessity of collaboration and confirms the algorithm’s ability to enable efficient information exchange.

**Weakness**

1.  The proposed method is relatively complex, incorporating multiple technical components such as subspace complementarity, model/structure similarity, distribution‑aligned synthetic graphs, and prototype‑based retrieval. While this design is principled, the number of interacting modules and hyperparameters may make the approach harder to tune and reproduce in practice.

2.  The prototype pool may gradually grow as new tasks and classes are added, potentially introducing storage overhead. While pruning can mitigate this, it relies on carefully chosen strategies and thresholds.

---

> ### Author Rebuttal · Authors · 2025-07-30
>
> We would like to thank you for acknowledging well-organization, sound logic, validity, and efficiency of proposed GHG. Thanks for your valuable suggestions!
>
> >W1: The proposed method is relatively complex. While this design is principled, the number of interacting modules and hyperparameters may make the approach harder to tune and reproduce in practice.
>
> A1: Our method introduces a practical graph socialized learning paradigm that leverages agent interactions to outperform individual agents. The modules are designed to answer the essential questions of socialized learning: with whom, what, and when to interact. It filters out collaborators’ low-quality, redundant, and insufficient information. Table 1 and Table 2 show that our approach incurs lower interaction costs than baselines and is insensitive to hyperparameters.
>
> * **Lower interaction costs.** The proposed method uses more interaction modules than baselines, yet achieves lower interaction costs and superior performance.
>
> **Table 1: Comparison of interaction memory size on CoraFull dataset.**
> | Method|Content|Memory size (MB) |MAP (%)|
> |:----|:----|:----|:----|
> |Fed-DMSG|Model parameters|1885.0|37.3|
> |POWER|Model parameters, Class prototype gradients|632.5|25.3|
> |MASC|Original graph, Classifiers, Node embeddings|109.4|36.7|
> |GHG|Prompts, Classifiers, Synthetic graphs, Node embeddings, Topology-aware embeddings |**105.0**|**54.0**|
>
>
> * **Insensitive hyperparameters.** The hyperparameters of our method are: the collaboration threshold **(analyzed in A2)**, the fine-grained collaborative weight $w_{col}$, the KL-divergence loss weight $w_{kl}$, and the synthetic standard-deviation loss weight $w_{\sigma}$. Table 2 shows that these are largely insensitive. Due to space limits, these results are placed in the appendix. Hyperparameter analysis aids tuning and reproduction. We will include it in the main text for the revised version.
>
> **Table 2: Hyperparameter analysis (MAP (\%)) on five datasets.**
> |$w_{col}$|0|0.2|0.4|0.6|0.8|1.0|1.2|1.4|1.6|
> |:----|:----|:----|:----|:----|:----|:----|:----|:----|:----|
> |CoraFull |49.0|50.5|52.4| 53.0| 53.1| 53.3| 53.1| 54.0| 53.4| 53.7|
> |Arxiv |56.3| 58.7| 59.0| 59.1| 59.2| 59.1| 59.2| 59.1| 59.1| 59.1|
> |Reddit |84.1| 86.0| 85.9| 86.1| 85.9| 86.2| 86.7| 86.1| 85.6| 85.7|
> |Cora |69.6| 79.2| 79.2| 79.1| 79.2| 79.1| 79.2| 79.0| 79.1| 79.2|
> |CiteSeer |59.4| 64.1| 64.8| 65.4| 65.7| 63.4| 63.6| 65.3| 63.8| 64.9|
>
> |$w_{kl}$|**1e-3**|**1e-2**|**1e-1**|**1e-0**| |$w_{\sigma}$|**1e-4**|**1e-3**|**1e-2**|**1e-1**|
> |:----|:----|:----|:----|:----|:----|:----|:----|:----|:----|:----|
> |CoraFull |51.5| 51.8| 51.9| 53.1|| |52.2|53.1|48.0|41.9|
> |Arxiv |59.0| 59.0| 59.2| 58.4||| 59.1|59.2|59.1|59.1|
> |Reddit |83.2| 83.7| 83.9| 86.7||| 86.5|86.7|86.1|85.7|
> |Cora |79.2| 78.6| 77.8| 74.2||| 78.5|78.8|79.2|75.9|
> |CiteSeer |56.2| 64.5| 62.3| 63.6|| |63.4|63.6|58.5|58.9|
>
>
>
>
> >W2: The prototype pool may gradually grow as new tasks and classes are added, potentially introducing storage overhead.
>
> >Q3: How sensitive is GHG to the choice of hyperparameters, such as the collaboration threshold or prototype pool size?
>
> A2: Each task prototype has a shape of $[1, d]$, so our method’s prototype pool is tiny and introduces negligible storage overhead. Thus, the prototype pool size is not treated as a hyperparameter. Table 3 shows that the collaboration threshold is largely insensitive.
>
> * **The prototype pool incurs negligible storage overhead when new tasks are added.** Each prototype is simply the mean of the received vectors, with dimension $[1, d]$. The entire pool occupies only $[T, d]$ after $T$ tasks, so we keep it unchanged. We believe that pruning might not be necessary.
>
> * **Low sensitivity of the collaboration threshold.** Table 3 summarizes the results under the adjusted thresholds: the vertical axis shows the complementary collaboration weight $w_c$ and the horizontal axis the similarity collaboration weight $w_s$. It can be observed that both thresholds are of low sensitivity.
>
> **Table 3: Collaboration threshold analysis (MAP (\%)) on CoraFull dataset.**
> |$w_c$/$w_s$|0.1|0.3|0.5|0.7|0.9|
> |:----|:----|:----|:----|:----|:----|
> |0.5|51.0|51.7|50.7|51.2|52.1|
> |0.7|51.9|51.8|51.0|51.0|51.0|
> |0.9|51.4|51.2|51.9|51.9|51.5|
> |1.1|51.4|51.8|52.4|52.0|51.8|
> |1.3|51.7|52.9|51.7|51.9|52.0|
> |1.5|51.9|51.7|52.3|52.2|52.0|
> |1.7|51.9|52.0|52.0|52.3|51.8|
>
>
>
>
>
>
> >Q1: The paper states that “D-GFL fails to capture agent dependencies,” but recent variants (e.g., PushSum-GNN) incorporate structure-aware designs. Could the authors clarify whether this refers only to standard D-GFL methods?
>
> A3: We clarify that "D-GFL fails to capture agent dependencies" refers to standard D-GFL methods. These methods ignore the special design of the communication topology, and the recent topology-aware variants (e.g., PushSum-GNN) cannot be directly transferred to graph data.
>
> * **Lack of dedicated designs for the communication topology in standard D-GFL approaches.**  D-GFL denotes classic methods (e.g., [1][2][3]), which rely solely on an undirected, symmetric topology that cannot meet individual agents’ collaboration needs.
>
>
> * **Inapplicability of recent variants (e.g., PushSum-GNN) to graph data.** Early structure-aware designs in vision attempt to build topologies from gradients, but the learned structures are task-specific, non-shareable of structure information, and disregard similarity or complementarity among agents, so they may still miss accurate agent dependencies.
>
> [1] Semi-decentralized federated ego graph learning for recommendation. WWW, 2023.
>
> [2] Spreadgnn: Decentralized multi-task federated learning for graph neural networks on molecular data. AAAI, 2022.
>
> [3] Decentralized federated graph neural networks. IJCAI Workshop, 2021.
>
>
>
> >Q2: Would it be beneficial to explore additional combinations—such as testing pairs of modules—to better understand their joint contributions and potential redundancy?
>
> A4: We evaluate the contributions of three module pairs across five datasets, replacing $L_{syn}$ with random sampling when $L_{syn}$ is omitted. Table 4 shows that every pair improves the model. Removing both $L_{ce}$ and $L_{kl}$ hurts most, confirming the necessity of interaction. Performance drops more with w/o $L_{syn} + L_{ce}$ than w/o $L_ {syn} + L_{kl}$, especially on Arxiv and Cora, underscoring the importance of task-specific information.
>
> **Table 4: Ablation comparisons (MAP (\%)) on five datasets.**
> |Datasets |CoraFull|Arxiv|Reddit|Cora|Citeseer|
> |:----|:----|:----|:----|:----|:----|
> |w/o $L_{ce}+L_{kl}$|33.2±0.3|26.4±0.3|48.3±0.2|54.1±0.2|52.6±0.7|
> |w/o $L_{syn}+L_{ce}$|45.8±0.3|30.0±0.4|71.4±0.2|53.7±1.7|53.8±2.8|
> |w/o $L_{syn}+L_{kl}$|47.8±0.7|56.2±0.7|82.8±0.2|70.0±2.8|56.3±1.7|
> |Full|54.0±0.8|59.2±0.3|86.7±0.9|79.2±3.4|65.7±4.1|

---

### Note · Authors · 2025-08-12

We would like to express our gratitude to all reviewers for their valuable feedback and discussion.

**We appreciate reviewers’ recognition of clear organization [#4UrH, #sYbT], strong motivation [#4UrH], novelty [#S53e, #4UrH], practicality [#4UrH], effectiveness [#S53e, #sYbT], efficiency [#sYbT], empirical rigor [#S53e, #sYbT], and competitive performance [#S53e, #4UrH] of our work.**

We have summarized concerns and responses as follows:

**Reviewer S53e** appreciated more comprehensive explanations and extended experiments. We have conducted detailed analyses on scalability, privacy guarantees, and memory footprint. In addition, we have performed experiments to validate the effectiveness of prototype mechanism in complex scenarios and strong performance of GHG across more diverse datasets.

**Reviewer 4UrH** responded positively to our detailed analysis and expanded experimental evidence, which led them to raise their score. We have demonstrated advantages of GHG over baselines, including shorter runtime, lighter-weight models, no need for replay, lower memory usage, and reduced interaction cost, and further validated the robustness of prototype mechanism in complex scenarios.

**Reviewer LUDz** acknowledged that our explanations regarding the use of benchmark datasets and data partitioning addressed their concerns. We have clarified the concept of completely new information. We have discussed interrelationship between agent numbers and performance under two scenarios: (1) the number of participating agents varies, and (2) the total number of agents varies (including an analysis of performance differences in strong/weak heterogeneous situations). We have presented detailed training and testing procedures (Algorithms 1 and 2 of appendix) and provided a thorough explanation regarding the case where forgetting rate is 0.

**Reviewer sYbT** provided positive comments on our work. We have demonstrated that GHG achieves lower interaction costs compared to baselines and exhibits low sensitivity to hyperparameters (Figure 3 of the main paper and Figures 7-8 of appendix). We have shown that prototype pool incurs negligible storage overhead and clarified that D-GFL fails to capture agent dependencies. We have added an ablation analysis on pairs of modules.

We sincerely appreciate the opportunity to revise paper and welcome further discussions to enhance its impact. We sincerely thank Area Chair for fostering fair, thorough, and balanced evaluation of our work.

---

### Decision · Program_Chairs · 2025-09-17

**Decision:**

Accept (poster)

**Comment:**

This paper proposes a graph socialized learning paradigm, instantiated as the "Graphs Help Graphs" method, to address multi-agent graph learning in heterogeneous and dynamic environments. The framework formalizes who to collaborate with, what to share, and when to communicate in a unified Bayesian paradigm. Key components include graph-driven organizational structures, synthetic subgraph exchange, and prototype-based life cycle management.

**Strengths:**
The paper is well-written, logically organized, and presents a novel and well-motivated paradigm for multi-agent graph learning. The idea of socialized collaboration (who/what/when) is intuitive, and the design integrates multiple techniques in a principled Bayesian way. Experiments are extensive, with ablations and heterogeneity sweeps, showing strong improvements over federated, lifelong, and hybrid baselines. The framework convincingly addresses both catastrophic forgetting and graph heterogeneity.

**Weaknesses and Suggestions:**
1. Despite its contributions, the approach is relatively complex, involving multiple modules and hyperparameters, which may raise reproducibility and scalability concerns.
2. The proposed formulation is closely related to DeLAMA [38]. A more thorough discussion of the differences and contributions is needed in the final version.
3. In the original version, the experiments are limited to citation/social-network benchmarks. Results on real-world applications (as provided in the rebuttal) should be incorporated into the final version.
4. The original paper lacks analysis of efficiency in terms of time, memory, and communication. The corresponding discussions from the rebuttal should also be included in the final version.

**Conclusion:**
Overall, this is a novel and impactful contribution to multi-agent graph learning, addressing both theoretical and practical challenges. While some limitations remain, they do not overshadow the significance of the work. Therefore, I recommend acceptance.